# Sequence Approximation using Feedforward Spiking Neural Network for Spatiotemporal Learning: Theory and Optimization Methods

**Xueyuan She, Saurabh Dash & Saibal Mukhopadhyay**
School of Electrical and Computer Engineering
Georgia Institute of Technology, Atlanta, GA 30332, USA
{xshe6,sdash38}@gatech.edu, saibal.mukhopadhyay@ece.gatech.edu

## Abstract

A dynamical system of spiking neurons with only feedforward connections can classify spatiotemporal patterns without recurrent connections. However, the theoretical construct of a feedforward spiking neural network (SNN) for approximating a temporal sequence remains unclear, making it challenging to optimize SNN architectures for learning complex spatiotemporal patterns. In this work, we establish a theoretical framework to understand and improve sequence approximation using a feedforward SNN. Our framework shows that a feedforward SNN with one neuron per layer and skip-layer connections can approximate the mapping function between any arbitrary pairs of input and output spike train on a compact domain. Moreover, we prove that heterogeneous neurons with varying dynamics and skip-layer connections improve sequence approximation using feedforward SNN. Consequently, we propose SNN architectures incorporating the preceding constructs that are trained using supervised backpropagation-through-time (BPTT) and unsupervised spiking-timing-dependent plasticity (STDP) algorithms for classification of spatiotemporal data. A dual-search-space Bayesian optimization method is developed to optimize architecture and parameters of the proposed SNN with heterogeneous neuron dynamics and skip-layer connections.

## 1 Introduction

Spiking neural network (SNN) (Ponulak & Kasinski, 2011) uses biologically inspired neurons and synaptic connections trainable with either biological learning rules such as spike-timing-dependent plasticity (STDP) (Gerstner & Kistler, 2002) or statistical training algorithms such as backpropagation-through-time (BPTT) (Werbos, 1990). The SNNs with simple leaky integrate-and-fire (LIF) neurons and supervised training have shown classification performance similar to deep neural networks (DNN) while being energy efficient (Kim et al., 2020b; Wu et al., 2019; Srinivasan & Roy, 2019). One of SNN's main difference from DNN is that the neurons are dynamical systems with internal states evolving over time, making it possible for SNN to learn temporal patterns without recurrent connections. Empirical results on feedforward-only SNN models show good performance for spatiotemporal data classification, using either supervised training (Lee et al., 2016; Kaiser et al., 2020; Khoei et al., 2020), or unsupervised learning (She et al., 2021). However, while empirical results are promising, a lack of theoretical understanding of sequence approximation using SNN makes it challenging to optimize performance on complex spatiotemporal datasets.

In this work, we develop a theoretical framework for analyzing and improving sequence approximation using feedforward SNN. We view a feedforward connection of spiking neurons as a spike propagation path, hereafter referred to as a *memory pathways* (She et al., 2021), that maps an input spike train with an arbitrary frequency to an output spike train with a target frequency. Consequently, we argue that an SNN with many memory pathways can approximate a temporal sequence of spike trains with time-varying unknown frequencies using a series of pre-defined output spike trains with known frequencies. Our theoretical framework aims to first establish SNN's ability to map frequencies of input/output spike trains within arbitrarily small error; and next, derive the basic principles for adapting neuron dynamics and SNN architecture to improve sequence approximation.

The theoretical derivations are then investigated with experimental studies on feedforward SNN for spatiotemporal classifications. We adopt the basic design principles for improving sequence approximation to optimize SNN architectures and study whether these networks can be trained to improve performance for spatiotemporal classification tasks. The key contributions of this work are:

- We prove that any spike-sequence-to-spike-sequence mapping functions on a compact domain can be approximated by feedforward SNN with one neuron per layer using skip-layer connections, which cannot be achieved if no skip-layer connection is used.

- We prove that using heterogeneous neurons having different dynamics and skip-layer connection increases the number of memory pathways a feedforward SNN can achieve and hence, improves SNN's capability to represent arbitrary sequences.

- We develop complex SNN architectures using the preceding theoretical observations and experimentally demonstrate that they can be trained with supervised BPTT and unsupervised STDP for spatiotemporal data classification.

- We design a dual-search-space option for Bayesian optimization process to sequentially optimize network architectures and neuron dynamics of a feedforward SNN considering heterogeneity and skip-layer connection to improve learning and classification of spatiotemporal patterns.

We experimentally demonstrate that our network design principles coupled with the dual-search-space Bayesian optimization improve classification performance on DVS Gesture (Amir et al., 2017), N-caltech (Orchard et al., 2015), and sequential MNIST. Results show that the design principles derived using our theoretical framework for sequence approximation can improve spatiotemporal classification performance of SNN.

## 2 RELATED WORK

Prior theoretical approaches to analyze SNN often focus on the storage and retrieval of precise spike patterns (Amit & Huang, 2010; Brea et al., 2013). There are also works that consider SNN for solving optimization problems (Chou et al., 2018; Binas et al., 2016) and works that analyze the dynamics of SNN (Zhang et al., 2019; Barrett et al., 2013). Those are different topics from the approximation of spike-sequence-to-spike-sequence mappings functions. SNN that incorporates excitatory and inhibitory signal is shown for its ability to emulate sigmoidal networks (Maass, 1997) and is theoretically capable of universal function approximation. Feedforward SNN with specially designed spiking neuron models (Iannella & Back, 2001; Torikai et al., 2008) have been demonstrated for function approximation, while for networks using LIF neurons, function approximation has been shown with only empirical results (Farsa et al., 2015). On the other hand, existing works that have developed efficient training process for SNN and demonstrated classification performance comparable to deep learning models, have mostly used simpler and generic LIF neuron models (Lee et al., 2016; Kaiser et al., 2020; Kim et al., 2020b; Wu et al., 2019; Sengupta et al., 2019; Safa et al., 2021; Han et al., 2020). Therefore, this paper develops the theoretical basis for function approximation using feedforward SNN with LIF neurons, and studies applications of the developed theoretical constructs in improving SNN-based spatiotemporal pattern classification.

The effectiveness of heterogeneous neurons (She et al., 2021) and skip-layer connections (Srinivasan & Roy, 2019; Sengupta et al., 2019) in SNN has been empirically studied in the past. However, no theoretical approach has been presented to understand why such methods improve learning of spike sequences, and how to optimize SNN's architecture and parameters to effectively exploit these design constructs. It is possible to search for the optimal SNN configurations through optimization algorithms, but the large amount of hyper-parameters for spiking neurons and network structure creates a high-dimensional search space that is long and difficult to solve. Bayesian optimization (Snoek et al., 2012) uses collected data points to make decision on the next test point that could provide improvement, thus accelerates the optimization process. Prior works (Parsa et al., 2019; Kim et al., 2020a) have shown that SNN performance can be effectively improved with Bayesian optimization. While those works consider a single or a few neuron parameters, the dual-search-space Bayesian optimization proposed in this work optimizes both network architecture and neuron parameters efficiently by separating the discrete search spaces from the continuous search spaces.

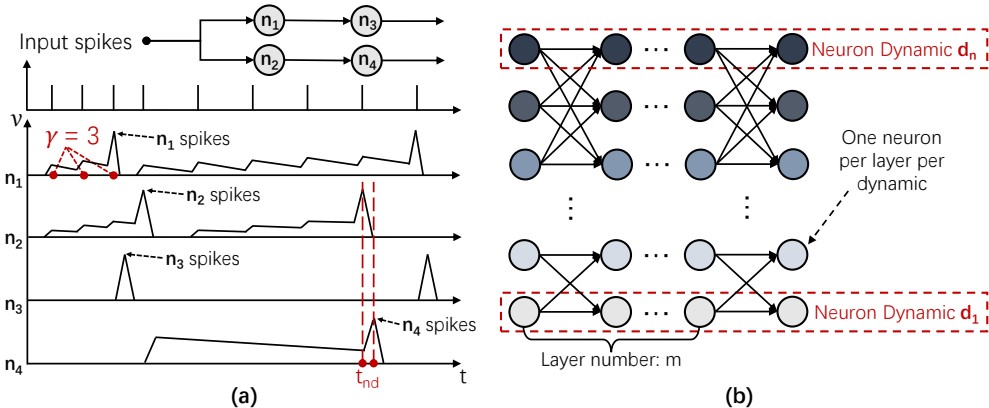

Figure 1: (a) A time-varying input spike sequence received by two memory pathways: neuron membrane potential plots show the different response from the neurons to the given input. (b) A minimal multi-neuron-dynamic (mMND) network with $m$ layers and $n$ neuron dynamics.

## 3 APPROXIMATION THEORY OF FEEDFORWARD SNN

### 3.1 DEFINITIONS AND NOTATIONS

**Definition 1** *Neuron Response Rate $\gamma$* For a spiking neuron $n$ with membrane potential at $v_{reset}$ and input spike sequence with period $t_{in}$, $\gamma$ is the number of input spike $n$ needs to reach $v_{th}$.

**Definition 2** *Memory Pathways* For a feedforward SNN with $m$ layers, a memory pathway is defined as a spike propagation path from input to the output layer. Two memory pathways are considered distinct if the set of neurons contained in them is different.

**Definition 3** *Minimal Multi-neuron-dynamic (mMND) Network* A densely connected network in which each layer has an arbitrary number of neurons that have different neuron parameters. All synapses from one pre-synaptic neuron have the same synaptic conductance.

**Notations** Neuron Delay $t_{nd}$ is the time for a spike from pre-synaptic neuron to arrive at its post-synaptic neurons, as shown in Figure 1(a). For a feedforward SNN with $m$ layers, a skip-layer connection can be defined with source layer and target layer pair $(l_s, l_t)$. The output feature map from source layer is concatenated to the original input feature map of the target layer. For the analysis of spike sequence in temporal space, the notation of $T_{max}$ and $T_{min}$ are defined as positive real numbers such that $T_{max} > T_{min}$. $\epsilon > 0$ is the error of approximation.

Figure 1(a) shows two memory pathways receiving an input spike sequence with time-varying periods. As the neurons have different dynamics, the two memory pathways have different response to the input spike sequence. An example of mMND network with $m$ layers and $n$ neuron dynamics is shown in Figure 1(b). SNN with multilayer perceptron (MLP) structure can be considered a scaled-up mMND network with multiple neurons for each dynamic. A network with convolutional structure can be considered a scaled-up mMND network with duplicated connections in each layer. We analyze the correlation of network capacity and structure based on mMND networks. The design of neuron heterogeneity can also be implemented in MLP-SNN and Conv-SNN as described in Section 4. The analysis for network capacity can be extended to those networks according to their specific layer dimensions.

### 3.2 MODELING OF SPIKING NEURON

SNN consists of spiking neurons connected by synapses. The spiking neuron model studied in this work is leaky integrate-and-fire (LIF) as defined by the following equations:

$$\tau_m \frac{dv}{dt} = a + R_m I - v; \ v = v_{reset}, \ \text{if } v > v_{threshold} \tag{1}$$

$R_m$ is membrane resistance, $\tau_m = R_m C_m$ is time constant and $C_m$ is membrane capacitance. $a$ is the resting potential. $I$ is the sum of current from all input synapses that connect to the neuron. A spike is generated when membrane potential $v$ cross threshold and the neuron enters refractory period $r$, during which the neuron maintains its membrane potential at $v_{reset}$. The time it takes for a pre-synaptic neuron to send a spike to its post-synaptic neurons is $t_{nd}$. Neuron response rate $\gamma$ is a property of a spiking neuron's response to certain input spike sequence. We show how the value of $\gamma$ can be evaluated below.

**Remark** For any input spike sequence, each individual spike can be described with Dirac delta function $\delta(t - t^i)$ where $t^i$ is the time of the $i$-th input spike. Consider membrane potential of a spiking neuron receiving the input before reaching spiking threshold, with initial state at $t = 0$ with $v = v_{reset}$, solving the differential equation (1) leads to (Gerstner, 1995):

$$v(t) = v_{reset}e^{-\frac{t}{\tau_m}} + a(1 - e^{-\frac{t}{\tau_m}}) + \frac{R_m}{\tau_m}e^{-\frac{t}{\tau_m}}\sum_i G \int_0^t \delta(t - t^i)e^{\frac{t}{\tau_m}}\,dt \qquad (2)$$

Here, $G$ is the conductance of input synapse connected to the neuron, which is trainable. From (2), there exists a value of $u$ such that $v_m(t^{u-1}) < v_{threshold}$ and $v_m(t^u) >= v_{threshold}$. By evaluating (2) for $u$ given neuron parameters and input spike sequence, the neuron response rate $\gamma$ can be found.

### 3.3 Approximation Theorem of Feedforward SNN

To develop the approximation theorem for feedforward SNN, we first aim to understand the range of neuron response rate that can be achieved. We show with Lemma 1 that for any input spike sequence with periods in a closed interval, it is possible to set the neuron response rate $\gamma$ to any positive integer. Based on this property, we show with Theorem 1 that by connecting a list of spiking neurons with certain $\gamma$ sequentially and inserting skip-layer connections, network with spike period mapping function $P(t)$ can be achieved to approximate the target spike sequence. To understand whether this capability of feedforward SNN relies on skip-layer connections, we develop Lemma 2 to prove that skip-layer connections are indeed necessary. In subsection 3.4 we investigate the correlation between approximation capability and network structures by analyzing the cutoff property of spiking neurons, which can change the network's connectivity. In our analysis, we focus on two particular designs: heterogeneous network (Lemma 4) and skip-layer connection (Lemma 5), and show their impact on the number of distinct memory pathways in a network. All lemmas are formally proved in the appendix.

**Lemma 1** *For any input spike sequence with period $t_{in}$ in range $[T_{min}, T_{max}]$, there exists a spiking neuron $n$ with fixed parameters $v_{th}, v_{reset}, a, R_m$ and $\tau_m$, such that by changing synaptic conductance $G$, it is possible to set the neuron response rate $\gamma_n$ to be any positive integer.*

*Proof Sketch.* (Formal proof in Appendix A) Given an input spike sequence, the highest possible membrane potential decay $\Delta v$ for any input $t_{in} \in [T_{min}, T_{max}]$ can be derived as a function of neuron parameters. We show it is possible to make $\Delta v$ tend to zero by setting the neuron parameters. Since the decay of $v$ can be negligible, $\gamma_n$ can be set to any positive integer by changing $G$.

**Theorem 1** *For any input and target output spike sequence pair with periods $(t_{in}, t_{out}) \in [T_{min}, T_{max}] \times [T_{min}, T_{max}]$, there exists a minimal-layer-size network with skip-layer connections that has memory pathway with output spike period function $P(t)$ such that $|P(t_{in}) - t_{out}| < \epsilon$.*

*Proof Sketch.* (Formal proof in Appendix B) With skip-layer connections, there can be multiple memory pathways in a minimal-layer-size network as neurons can be either included or skipped through. Hence it is possible to create memory pathways with different delay times for each input spike in a network. By connecting the output of those memory pathways to a common neuron $n'$, spike sequence of any arbitrary period $t_{int}$ such that $t_{int} <= t_{in}$ can be generated within $\epsilon$. By setting $\gamma_{n'} > 1$, the output from $n'$ receiving input spike sequence with $t_{int}$ is $t_{out}^{n'} = \gamma \cdot t_{int}$. Hence it is possible to achieve a network with output spike period $P(t)$ such that $|P(t_{in}) - t_{out}| < \epsilon$.

**Lemma 2** *With no skip-layer connection, there does not exist a minimal-layer-size network that has output spike period function $P(t)$ such that for any input and target output spike sequence pair with periods $(t_{in}, t_{out}) \in [T_{min}, T_{max}] \times [T_{min}, T_{max}]$, $|P(t_{in}) - t_{out}| < \epsilon$.*

*Proof Sketch.* (Formal proof in Appendix C) A minimal-layer-size network without skip-layer connection has only one memory pathway. For a particular input spike sequence with period $t_{in}$, different output period $P(t_{in})$ can be achieved by changing $\gamma$ of neurons in the memory pathway. We show that there exists two output spike periods $P(t_{in})$ and $P'(t_{in})$, such that $P(t_{in}) - P'(t_{in})$ is a constant value independent of network or neuron configurations, and there can be no $P''(t_{in})$ such that $P(t_{in}) < P''(t_{in}) < P'(t_{in})$. Therefore, for any minimal-layer-size network, there exists $t_{out}$ within the range of $(P(t_{in}), P'(t_{in}))$ such that $|P(t_{in}) - t_{out}| < \epsilon$ does not hold true.

## 3.4 NETWORK STRUCTURE AND MEMORY PATHWAYS

Based on Theorem 1, it is possible to approximate an input/output spike sequence mapping function using a minimal-layer-size network with specific configuration, which can be considered as a memory pathway. Since any bounded continuous function on a compact domain can be approximated to arbitrary accuracy using a piece-wise constant function, and it is possible to use a memory pathway to approximate each of the constant function, with increasing number of distinct memory pathways, a feedforward SNN can achieve approximation of continuous functions with less error. In this subsection, we show that with two structural designs: heterogeneous network i.e. a network having neurons with different dynamics and adding skip-layer connections, a feedforward SNN has the capability to achieve more distinct memory pathways.

**Cutoff Frequency of a Memory Path**   We first show the correlation of cutoff period and spiking neuron parameters with Lemma 3, which is proved in Appendix D.

**Lemma 3** *A spiking neuron has cutoff period $\omega_c = \tau_m \ln\left(\frac{v_{reset} - a}{v_{reset} - a + \frac{R_m}{\tau_m} G}\right)$ above which input spike sequence cannot cause the spiking neuron to spike.*

**Remark** From Lemma 3, it can be observed that the cutoff period $\omega_c$ of a neuron can be configured to any positive real number by changing the neuron parameters and synaptic conductance $G$. Further, with fixed $G$, $\omega_c$ can be configured to any positive real number by changing the neuron parameters. Neurons that are in cutoff change the spike propagation path in a network as they send no output spikes. This creates different memory pathways without changing the connections in a network.

**Heterogeneous Network**   If an mMND network has the same parameters for all neurons in each layer, the majority of the neurons are included in the same memory pathway, leading to the upper bound of number of distinct memory pathways to be limited. With Lemma 4, we show the relationship between the upper bound of the number of distinct memory pathways and the number of different neuron dynamics in an mMND network.

**Lemma 4** *For an mMND network with $m$ layers and $\{\lambda_1, \lambda_2, ...\lambda_m\}$ number of different neuron dynamics in each layer, the least upper bound of the number of distinct memory pathways is $\prod_{i=1}^{m} \lambda_i$.*

*Proof Sketch.* (Formal proof in Appendix D) For an mMND network, it is possible to have neurons with different $\omega_c$ in each layer, which creates $\lambda_i$ number of different neuron activation states for layer $i$. Across all network layers, the highest possible number of different neuron activation states is therefore the product of $\lambda$ of each layer. Since neurons in cutoff do not propagate spikes, they can be removed from a memory pathway. This leads to $\prod_{i=1}^{m} \lambda_i$ as the least upper bound of the number of memory pathways.

Compared to a network with homogeneous neuron parameters, in which the upper bound of number of distinct memory pathways is $\lambda_m$, Lemma 4 indicates that heterogeneous network increases the maximum achievable number of distinct memory pathways in a feedforward SNN.

**Skip-layer Connection**   We show that adding skip-layer connection increases the upper bound of the number of memory pathways in a network with Lemma 5.

**Lemma 5** *For an mMND network with $m$ layers and $\{\lambda_1, \lambda_2, ...\lambda_m\}$ different neuron dynamics in each layer and a skip-layer connection made between layer $l_a$ and $l_b$, s.t. $a, b \in \{1, 2, ...m\}$ and $(b - a) > 1$, the least upper bound of the number of memory pathways is $\prod_{i=1}^{m} \lambda_i + (\prod_{i=1}^{a} \lambda_i \cdot \prod_{i=b}^{m} \lambda_i)$*

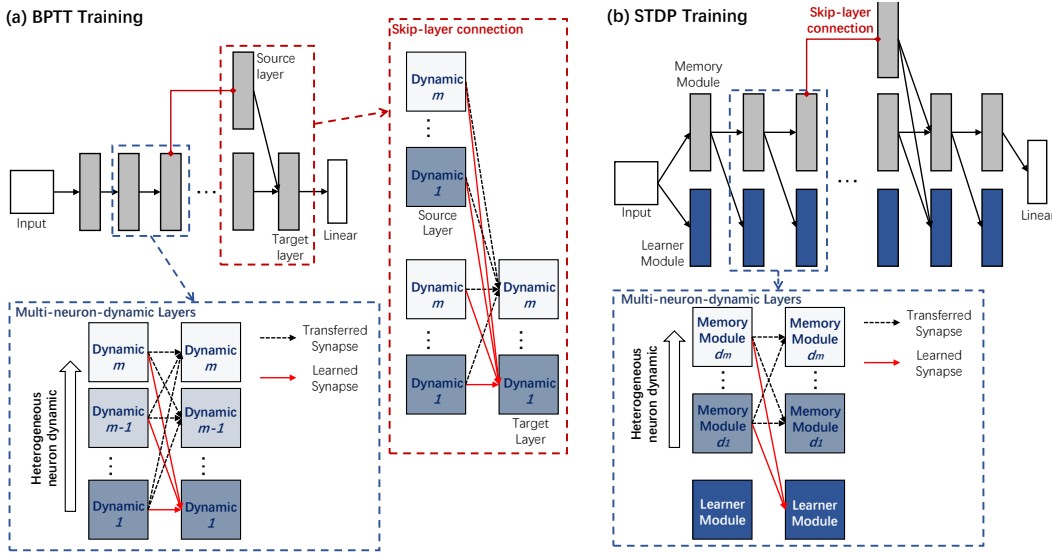

Figure 2: (a) The proposed network with BPTT training, each multi-neuron-dynamic layer contains a set of neuron dynamics from $d_1$ to $d_m$. (b) The proposed network with STDP training.

*Proof Sketch.* (Formal proof in Appendix D) Compared to the network considered in Lemma 4, by adding a skip-layer connection, there are additional possible neuron activation states in the network that result from the cutoff of neurons in layers between $l_a$ and $l_b$. Without layers between $l_a$ and $l_b$ in the spike propagation path, the least upper bound of the number of memory pathways is increased by the maximum number of distinct memory pathways in a network that has layers $\{l_1, l_2, ..., l_a, l_b, l_{b+1}, ..., l_m\}$ connected sequentially.

## 4 SNN ARCHITECTURE DESIGN USING APPROXIMATION THEORY

In this section, we discuss design of SNN architectures as inspired by the developed approximation theory for feedforward SNN, with more details in Appendix G.

**Network Template for BPTT Training** For BPTT training, the network template is shown in Figure 2(a). The heterogeneity in neuron dynamics is implemented by using the multi-neuron-dynamic layers, which can either be convolutional or fully connected. The multi-neuron-dynamic layers use different neuron parameters for spiking neurons in each neuron dynamic module, which contains a certain number of feature maps for convolutional layers or a certain number of neurons for fully-connected layers. There are two types of synapses between layers: transferred synapses marked as black dashed arrows and learned synapses marked as red solid arrows. The conductance of learned synapses is optimized by the BPTT algorithm during training, and the transferred synapses have the same conductance as the learned synapses from the same pre-synaptic neuron. During forward pass, neurons in each layer receive the same input features and respond differently based on their neuron dynamics to generate different output features. During back-propagation, only conductance of the learned synapses are updated. Skip-layer connection is implemented with the output spike matrix from source layer concatenated to the original input spike matrix of the target layer. The skip-layer connection has the same implementation as the regular connection between consecutive layers, with both learned and transferred synapses (Figure 2(a)). The last layer of the network is a fully-connected layer with homogeneous dynamic to generate prediction labels.

**Network Template for STDP Learning** For networks trained with STDP, the template is shown in Figure 2(b). Each layer contains a learner module and a memory module. Learner modules use homogeneous neuron dynamic that is suitable for STDP learning, and memory modules consist of neurons with different dynamics. There are also two types of synapses: transferred synapses and learned synapses. Between two layers, memory modules are connected with transferred synapses and memory modules are connected to learner modules with learned synapses. Leaner modules be-

tween layers are not directly connected. STDP training proceeds as a layer-by-layer process. During training of the first layer, conductance of synapses connecting neurons in memory module to neurons in learner module is learned with STDP using all training data without supervision. Then, the learned conductance is transferred to the transferred synapses in the same layer. During training of the second layer, layer 1 memory module perceives the same input features and generates different output features with the heterogeneous neurons. This lay-by-layer process is repeated until the layer before the final layer finishes learning. The final linear layer is then fine-tuned using stochastic gradient descent (SGD) based on spike frequency array from the last multi-neuron-dynamic layer generated based on the labeled data. Skip-layer connection is implemented by connecting the memory module of the source layer to the target layer. The connections are made with the two types of synapses and follow the same training process as the consecutive layers.

**Dual-search-space Bayesian Optimization** Bayesian optimization uses Gaussian process to model the distribution of an objective function, and an acquisition function to decide points to evaluate. For data points in a target dataset $x \in X$ and the corresponding label $y \in Y$, an SNN with network structure $\mathcal{V}$ and neuron parameters $\mathcal{W}$ acts as a function $f_{\mathcal{V},\mathcal{W}}(x)$ that maps input data $x$ to predicted label $\tilde{y}$. The optimization problem in this work is defined as

$$\min_{\mathcal{V},\mathcal{W}} \mathcal{P} \quad \text{where} \quad \mathcal{P} = \sum_{x \in X, y \in Y} \mathcal{L}(y, f_{\mathcal{V},\mathcal{W}}(x)) \tag{3}$$

$\mathcal{V}$ contains the number of layers $N_{layers}$, the number of memory dynamics $N_{dynmaic}$ and skip-layer connection configuration variables $N_{skip}$, $L_{start}$ and $L_{end}$, each controlling the number of skip-layer connections, the first layer and last layer to implement skip-layer connections. All of the values are discrete. $\mathcal{W}$ contains the values for $a$, $\tau_m$ and $R_m$ in (1), which are continuous. We separate the discrete and continuous search spaces by implementing a dual-search-space optimization process, where $\mathcal{V}$ is first optimized with fixed, manually tuned neuron parameters. After an optimal structure is found, $\mathcal{W}$ are optimized for the selected $\mathcal{V}$. Details on the configurations of the optimization process are listed in the appendix. To achieve Bayesian optimization with constraints, we implement a modified expected improvement (EI) acquisition function similar to the one shown by Gardner (Gardner et al., 2014), which uses a Gaussian process to model the feasibility indicator due to its high evaluation cost. In this work, since the constraint function can be explicitly defined, we use a feasibility indicator that is directly evaluated. The modified EI function is defined as:

$$I_c(\mathbb{W}) = \Delta(\mathbb{W}) \cdot \max\{0, \mathcal{P}(\mathbb{W}) - \mathcal{P}(\mathbb{W}^+)\} \tag{4}$$

where $\mathbb{W}$ is the network configuration containing $\mathcal{W}$ and $\mathcal{V}$. $\mathbb{W}^+$ is the test point that provided the best result. $\Delta(\mathbb{W})$ is the explicitly defined indicator function that takes the value of 1 when all constraints are satisfied and 0 otherwise.

## 5 EXPERIMENTS

### 5.1 EXPERIMENT SETTINGS

Datasets tested in the experiment include the DVS Gesture (Amir et al., 2017), which is a human gesture dataset captured by DVS cameras, and the N-Caltech101 (Orchard et al., 2015), which is an event-based version of the Caltech101 dataset. The proposed method is also tested for MLP-style SNN on the sequential MNIST dataset presented row-by-row. We also vary the amount of *labeled data used during training* ranging from using 100% labeled data for training down to 10% labeled data (30% for N-Caltech101) during training. Note, during STDP training networks always use the entire but *un-labeled* training dataset; however, only the fraction of the labeled data is used for supervised fine-tuning of the last layer. Comparison is made for DVS Gesture and N-Caltech101 with prior works including ConvLSNN, which is a combination of convolutional SNN and recurrent SNN with long and short-term neurons trained with BPTT (Salaj et al., 2020), DECOLLE (Kaiser et al., 2020), which uses surrogate gradient to train a convolutional feedforward SNN, HATS (Sironi et al., 2018), which implements time surfaces and SVM for classification and H-SNN (She et al., 2021) which uses STDP to train a convolutional SNN with two neuron dynamics. Additional function approximation experiments are discussed in Appendix E.

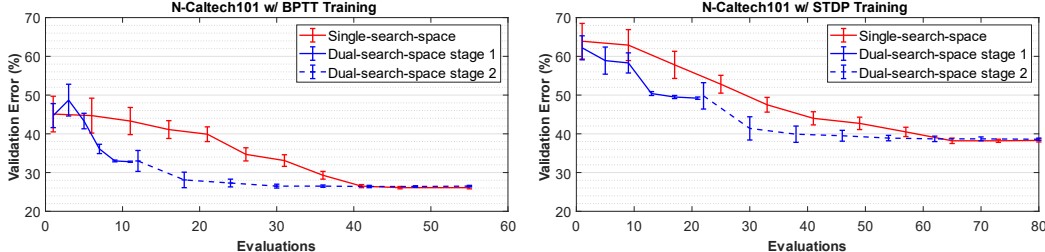

Figure 3: Validation error over optimization iterations for the proposed dual-search-space Bayesian optimization compared to the normal single-search-space Bayesian optimization.

## 5.2 EFFECT OF DUAL-SEARCH-SPACE BAYESIAN OPTIMIZATION

We compare the proposed dual-search-space Bayesian optimization with regular Bayesian optimization using a single search space for network validation error over 5 runs. The result from the N-Caltech101 dataset is shown in Figure 3. It can be observed that the two optimization approaches achieve similar minimum validation error after convergence. By separating the search spaces, the proposed optimization process reaches convergence faster than regular single-search-space optimization. It is also worth noting that, between the two stages in the optimization process for BPTT training, the first stage accounts for more reduction in validation error than the second stage. This indicates that optimizing network structure causes more impact to BPTT training than optimizing neuron parameters, which is potentially due to the reason that network structure more heavily affects the number of memory pathways in the network than neuron parameters. On the other hand, for STDP training where learning behavior is sensitive to the dynamic of spiking neurons, the reduction of validation error is more equally shared between the two optimization stages. Over the 5 runs, among all network configurations achieved after the dual-search-space optimization converges, we compare the configuration with the lowest number of trainable parameters against baseline models. The specific configurations for the optimized networks are listed in Table 1. It can be observed that for BPTT algorithm, the optimized networks have more layers than the STDP trained networks, and the optimal values found for neuron parameters are highly distinct for the two training methods.

Table 1: Configuration of optimized network models

| Network | Conv. Layer Number | Skip-layer Connection | Number of Different Neuron Dynamics and $a$ | Neuron Parameters $\tau_m$ | $R_m$ |
|---|---|---|---|---|---|
| BPTT, Gesture | 9 | (2,7) | 4, (-24,-17,-12,-9) | 120 | 340 |
| BPTT, N-Caltech | 12 | (2,5), (5,8), (8,11) | 5, (-23,-16,-14,-11,-8) | 70 | 300 |
| STDP, Gesture | 6 | (2,4), (4,6) | 4, (-26,-24,-15,-9) | 110 | 260 |
| STDP, N-Caltech | 8 | (3,5), (5,7) | 6, (-21,-19,-17,-13,-9,-7) | 140 | 240 |

## 5.3 ABLATION STUDIES

To investigate the effect of using multiple neuron dynamics, we apply the same dual-search-space Bayesian optimization process for networks that have homogeneous neuron dynamic for the same

Table 2: Ablation studies of optimized networks

| Model | Heterogeneity | Skip-layer | DVS Gesture | N-Caltech101 | S-MNIST |
|---|---|---|---|---|---|
| Homogeneous-BPTT | N | Y | 95.0 | 65.3 | 95.5 |
| No-skip-layer-BPTT | Y | N | 96.5 | 63.5 | 94.8 |
| **This Work-BPTT** | Y | Y | 98.0 | 71.2 | 97.3 |
| Homogeneous-STDP | N | Y | 91.3 | 37.0 | 94.3 |
| No-skip-layer-STDP | Y | N | 93.1 | 51.9 | 95.5 |
| **This Work-STDP** | Y | Y | 96.6 | 58.1 | 96.1 |

Table 3: Accuracy (%) for DVS Gesture (top) and N-Caltech101 (bottom)

| Model | Labeled Data % In Training | | | | Parameter Number |
|---|---|---|---|---|---|
| | 100% | 50% | 30% | 10% | |
| ConvLSNN (Salaj et al., 2020) | 97.1 | 95.3 | 92.0 | 84.3 | 2.9M |
| DECOLLE (Kaiser et al., 2020) | 97.5 | 95.0 | 91.2 | 83.9 | 1.3M |
| (Fang et al., 2021) | 97.8 | - | - | - | - |
| HATS (Sironi et al., 2018) | 95.2 | 94.1 | 91.6 | 83.7 | - |
| H-SNN (She et al., 2021) | 96.2 | 95.8 | 93.7 | 88.2 | 0.74M |
| **This Work-STDP Training** | 96.6 | **96.0** | **94.1** | **91.2** | 0.81M |
| **This Work-BPTT Training** | **98.0** | 95.3 | 91.1 | 82.4 | 1.1M |

| Model | Labeled Data % In Training | | | | Parameter Number |
|---|---|---|---|---|---|
| | 100% | 70% | 50% | 30% | |
| ConvLSNN (Salaj et al., 2020) | 63.1 | 58.7 | 51.3 | 45.4 | 3.0M |
| DECOLLE (Kaiser et al., 2020) | 66.9 | 61.9 | 56.2 | 50.6 | 2.0M |
| HATS (Sironi et al., 2018) | 64.2 | 61.0 | 54.3 | 48.8 | - |
| H-SNN (She et al., 2021) | 42.8 | 41.9 | 37.0 | 34.6 | 1.7M |
| **This Work-STDP Training** | 58.1 | 57.8 | **57.2** | **54.6** | 1.4M |
| **This Work-BPTT Training** | **71.2** | **65.4** | 56.0 | 52.5 | 1.7M |

number of evaluations as the proposed design. Similarly, to study the contribution to performance gain from skip-layer connections, the Bayesian optimization process is used for network templates without skip-layer connections. The optimization process runs for the same number of evaluations as the proposed design. From the results shown in Table 2, it can be observed that compared to baselines, the proposed networks achieve the best accuracy for all datasets. Specifically, when homogeneous network is used, the performance of STDP trained network is noticeably lower than the proposed method for DVS Gesture and N-Caltech101. For BPTT training, using heterogeneous network and skip-layer connection show different level of benefit for each dataset. For sequential MNIST which has less complexity, the improvement from using heterogeneous neurons and skip-layer connections is not as significant.

## 5.4 Comparison with Prior Works

**DVS Gesture** As shown in Table 3 (top), with 100% labels, the proposed network trained with BPTT demonstrates higher accuracy than all tested networks without using the most trainable parameters. The proposed network trained with STDP has slightly lower accuracy than some baselines when 100% labels are used; for reduced-label training it outperforms all other networks.

**N-Caltech101** As shown in Table 3 (bottom), the proposed network trained with BPTT outperforms all baselines with both 70% and 100% training labels and also has less trainable parameters than most baselines. The un-supervised learning models i.e., H-SNN and the proposed network with STDP, show considerably lower performance (more than what was observed for DVS Gesture) than supervised ones when 100% labels are available; However, the proposed network with STDP shows better performance than H-SNN, and outperforms all networks when available labels are below 50%.

## 6 Conclusion

We develop a theoretical basis to understand and optimize the ability of a feedforward SNN to approximate temporal sequence. We analytically show how heterogeneity and skip-layer connections can improve sequence approximation with SNN, and empirically demonstrate their impact on spatiotemporal learning. It is well-known in neuroscience that, heterogeneity (De Kloet & Reul, 1987) and irregular connectivity (Eickhoff et al., 2018) are intrinsic properties of human brains. Our analysis shows that incorporating such concepts within artificial SNN is beneficial for designing high-performance SNN for classification of spatiotemporal data.

ACKNOWLEDGMENTS

This material is based on work sponsored by the Army Research Office and was accomplished under Grant Number W911NF-19-1-0447. The views and conclusions contained in this document are those of the authors and should not be interpreted as representing the official policies, either expressed or implied, of the Army Research Office or the U.S. Government.

Authors would like to thank Drs. Mark Mclean, Chris Purdy, Fernando Camacho, and James Keiser from Laboratory for Physical Sciences for many helpful discussions on this work.

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

## A    SNN DYNAMICS

**Remark** For a sequentially connected neuron list with $m$ neurons all with $\gamma = 1$ and neuron delay $t_{nd}$, an input spike at time $t$ leads the neuron list to generate an output spike at time $t + mt_{nd}$

**Remark** For any input sequence with period $t_{in}$ to a spiking neuron with response rate $\gamma$ such that $\gamma > 1$, if refractory period is set to $r < t_{in}$, the neuron can exit refractory period before the next spike arrives.

**Lemma 1** For any input spike sequence with period $t_{in}$ in range $[T_{min}, T_{max}]$, there exists a spiking neuron $n$ with fixed parameters $v_{th}, v_{reset}, a, R_m$ and $\tau_m$, such that by changing synaptic conductance $G$, it is possible to set the neuron response rate $\gamma_n$ to be any positive integer.

*Proof.*    For a given input spike sequence period $t_{in}$, consider the maximum possible membrane potential decay that can be reached within a period of $t_{in}$. From (1), when $I = 0$, $\frac{dv}{dt} < 0$ and $\left|\frac{dv}{dt}\right|$ increases with higher $v$. Hence, the maximum decay of $v$ is reached when initial membrane potential $v(t = 0) \to v_{th}^-$ and the neuron decays for period $t_{in} = T_{max}$. The decayed membrane potential $v(t = T_{max})$ can be derived by solving the differential equation (1) for $v(t) = v_{th}$ at $t = 0$:

$$v(t = T_{max}) = v_{th}e^{-\frac{T_{max}}{\tau_m}} - ae^{-\frac{T_{max}}{\tau_m}} + a \tag{5}$$

It is possible to have a spiking neuron with $R_m, a$ and $\tau_m$ such that $\Delta v$, defined as

$$\Delta v = v(t = T_{max}) - v(t = 0) = v_{th}e^{-\frac{T_{max}}{\tau_m}} - ae^{-\frac{T_{max}}{\tau_m}} + a - v_{th} \tag{6}$$

,

tends to zero. With this configuration, since the the highest possible decay of membrane potential is negligible, for any target $\gamma$, it is possible to set $G$ such that

$$G = \frac{v_{th} - v_{reset}}{\gamma} \tag{7}$$

The proof is complete.

Note, input sequence that has no spike has $t_{in}$ tending to infinity, thus violating the bounded constraint mentioned in Lemma 1.

## B    PROOF OF THEOREM 1

**Theorem 1** For any input and target output spike sequence pair with periods $(t_{in}, t_{out}) \in [T_{min}, T_{max}] \times [T_{min}, T_{max}]$, there exists a minimal-layer-size network with skip-layer connections that has memory pathway with output spike period function $P(t_{in})$ such that $|P(t_{in}) - t_{out}| < \epsilon$.

*Proof.* For any given $(t_{in}, t_{out})$, first consider the condition where $t_{in} > t_{out}$. It is possible to construct a minimal-layer-size network $N$ connecting $m$ spiking neurons with neuron response rate $\gamma = 1$ sequentially, denoted as a $m$-tuple of neurons $\{n_1, n_2, ..., n_m\}$. Since any configuration of skip-layer connection with source layer and target layer pair $(l_s, l_t)$, such that $l_s \in [1, m - 2]$, and $l_t \in [l_s + 2, m]$, can be added, it is possible to add a $(m - 2)$-tuple of skip-layer connections

$$S_{sl} = \{(i, m) \ \forall \ i \in \{1, 2, 3, ..., m - 2\}\} \tag{8}$$

Denote the synaptic conductance for all the skip-layer connections as a $(m - 2)$-tuple

$$S_{G^{sl}} = \{G_1^{sl}, G_2^{sl}, G_3^{sl}, ..., G_{m-2}^{sl}\} \tag{9}$$

For any $t_{out} < t_{in}$, it is possible to find a $k$-tuple of synaptic conductance

$$S'_{G^{sl}} = \{G_i^{sl}, G_{2i}^{sl}, G_{3i}^{sl}, ..., G_{ki}^{sl}\} \ \text{s.t.} \ i = \lfloor\frac{t_{out}}{t_{nd}}\rfloor \ \text{and} \ k = \lfloor\frac{m-2}{i}\rfloor \tag{10}$$

Set synaptic conductance in $S_{G^{sl}} \setminus S'_{G^{sl}}$ to 0. Then set the conductance of synapse connecting $n_{m-1}$ and $n_m$ to 0. In such way, The output spikes from network N have period

$$P(t_{in}) = \lfloor \frac{t_{out}}{t_{nd}} \rfloor \cdot t_{nd} \tag{11}$$

For given $\epsilon$, it is possible to choose $t_{nd}$ such that $t_{nd} < 2\epsilon$, therefore satisfying $|P(t_{in}) - t_{out}| < \epsilon$. $m$ can be chosen as

$$m = \frac{T_{max} - T_{min}}{t_{nd}} \tag{12}$$

or equivalently:

$$m = \frac{T_{max} - T_{min}}{2\epsilon} \tag{13}$$

Since $\frac{T_{max} - T_{min}}{2\epsilon}$ is finite, $m$ is finite.

For $t_{in} < t_{out}$, using $N$ as described above, it is possible to achieve output spike with period within $\epsilon$ of any period in $(0, t_{in}]$. For a given $t_{out}$, assume the configuration in neuron list $N$ has output spike interval $t'_{int}$ such that $kt'_{int} = t_{out}$, where $k$ is a positive integer. From Lemma 1, it is possible to set $G$ for a neuron $n_{m+1}$ such that its neuron response delay satisfies $\gamma_{n_{m+1}} = k$ for input spike period $t'_{int}$. A new network, denoted as $N'$, can be formed by connecting $n_{m+1}$ to the output of $N$. $N'$ has output spike with period $P(t_{in}) = kt'_{int} = t_{out}$. Hence, to reach the given $\epsilon$, it requires neuron list $N$ to have output spike interval $t_{int}$ such that

$$|t_{int} - t'_{int}| < \frac{\epsilon}{k} \tag{14}$$

Since $k$ is finite, (14) can be achieved.

For $t_{in} >= t_{out}$, it is possible to configure network $N'$ such that $t_{int}$ satisfies $|t_{int} - t_{out}| < \epsilon$, and the value of $\gamma_{n_{m+1}}$ set to 1, hence $|P(t_{in}) - t_{out}| < \epsilon$ can be achieved. The proof is complete.

## C    PROOF OF LEMMA 2

**Lemma 2** With no skip-layer connection, there does not exist a minimal-layer-size network that has output spike period function $P(t_{in})$ such that for any input and target output spike sequence pair with periods $(t_{in}, t_{out}) \in [T_{min}, T_{max}] \times [T_{min}, T_{max}]$, $|P(t_{in}) - t_{out}| < \epsilon$.

*Proof.* A minimal-layer-size network $N$ with $m$ layers can be denoted as a $m$-tuple of neurons $\{n_1, n_2, ..., n_m\}$ connected sequentially. Since no skip-layer connection exists, there is only one distinct memory pathway that contains all neurons $\{n_1, n_2, ..., n_m\}$.

Denote the set of neuron response rate corresponding to each neuron in $N$ as

$$\Gamma = \{\gamma_1, \gamma_2, ..., \gamma_m\} \tag{15}$$

For a given input sequence with $t_{in}$, denote the timing of the first spike as $t_1^{in}$, consider the output spike sequence for network with

$$\gamma_i = 1 \ \forall \ \gamma_i \in \Gamma \tag{16}$$

The first output spike from $N$ has timing $\tilde{t}_1^{out} = t_1^{in} + mt_{nd}$, and the second output spike has timing $\tilde{t}_2^{out} = t_1^{in} + t_{in} + mt_{nd}$. It can be easily derived that the period of the output spike sequence is

$$P(t_{in}) = t_{in} \tag{17}$$

Also consider the output spike sequence for network with

$$\gamma_j = 2 \text{ for any } j \in \{1, 2, 3, 4, ..., m\} \text{ and } \gamma_i = 1 \ \forall \ i \in (\{1, 2, 3, 4, ..., m\} \setminus \{j\}) \tag{18}$$

Following the same process, the period of the output spike sequence is

$$P(t_{in}) = 2t_{in} \tag{19}$$

Since the smallest increase to any $\gamma_i$ is by 1, there is no set of values for $\gamma$ such that the network output spike sequence has period $P(t_{in})$ satisfying $t_{in} < P(t_{in}) < 2t_{in}$. Since within the range $(t_{in}, 2t_{in})$, there exists values of $t_{out}$ such that $|P(t_{in}) - t_{out}| < \epsilon$ does not hold. The proof is complete.

## D MEMORY PATHWAYS IN SNN

In this section we analyze the increase to the least upper bound of the number of memory pathways in a network by using heterogeneous networks and skip-layer connections.

**Lemma 3** A spiking neuron has cutoff period $\omega_c = \tau_m \ln(\frac{v_{reset} - a}{v_{reset} - a + \frac{R_m}{\tau_m}G})$ above which input spike sequence cannot cause the spiking neuron to spike.

*Proof.* Consider (2), since membrane potential increases at time $t^i$ and decays otherwise, solving for $t = t^i$ and the equation can be expanded:

$$v_m(t^i) = v_{reset}e^{\frac{t^i}{\tau_m}} + a(1 - e^{\frac{t^i}{\tau_m}}) + \frac{R_m}{\tau_m}Ge^{\frac{t^i - t^1}{\tau_m}} + \frac{R_m}{\tau_m}Ge^{\frac{t^i - t^2}{\tau_m}} + ... + \frac{R_m}{\tau_m}G \tag{20}$$

For input with frequency $f$, $t^{i+1} - t^i = \Delta t = \frac{1}{f}$, subtracting membrane potential values at two consecutive $t_i$ provides:

$$\Delta v_m = v_m(t^{i+1}) - v_m(t^i) = v_{reset}(e^{\frac{t^{i+1}}{\tau_m}} - e^{\frac{t^i}{\tau_m}}) - a(e^{\frac{t^{i+1}}{\tau_m}} - e^{\frac{t^i}{\tau_m}}) + \frac{R_m}{\tau_m}Ge^{\frac{t^{i+1} - t^1}{\tau_m}} \tag{21}$$

setting time of the first input spike $t^1$ to zero leads to:

$$\Delta v_m = e^{\frac{t^i}{\tau_m}}((e^{\frac{\Delta t}{\tau_m}} - 1)(v_{reset} - a) + \frac{R_m}{\tau_m}Ge^{\frac{\Delta t}{\tau_m}}) \tag{22}$$

As $e^{\frac{t^i}{\tau_m}} > 0$, and the term $((e^{\frac{\Delta t}{\tau_m}}) - 1)(v_{reset} - a) + \frac{R_m}{\tau_m}Ge^{\frac{\Delta t}{\tau_m}})$ does not depend on $t^i$, the polarity of $\Delta v_m$ does not change with time. $v_m$ is either strictly increasing, staying the same or decreasing with higher $t^i$. This indicates that, when $\Delta v_m \leq 0$ the post-synaptic neuron can never spike regardless of how many pre-synaptic spikes it receives. $\Delta v_m \leq 0$ when input spike period $t_{in}$ satisfies

$$t_{in} \geq \tau_m \ln(\frac{v_{reset} - a}{v_{reset} - a + \frac{R_m}{\tau_m}G}) \tag{23}$$

Therefore, the cutoff period of the neuron is

$$\omega_c = \tau_m \ln(\frac{v_{reset} - a}{v_{reset} - a + \frac{R_m}{\tau_m}G}) \tag{24}$$

The proof is complete.

In the following proof, we consider cutoff frequency, $f_c = \frac{1}{\omega_c}$ of spiking neurons. From (24) it can be easily derived that $f_c$ can be configured to any positive real number by changing the neuron parameters.

**Lemma 4** For an mMND network with $m$ layers and $\{\lambda_1, \lambda_2, ...\lambda_m\}$ number of different neuron dynamics in each layer, the least upper bound of the number of memory pathways is $\prod_{i=1}^{m} \lambda_i$.

*Proof.* Denote the set of neurons in layer $l$ with distinct neuron dynamics as

$$S_n^l = \{n_1^l, n_2^l, n_3^l, ..., n_{\lambda_l}^l\} \tag{25}$$

Since the network is mMND, $|S_n^l| = \lambda_l$. Denote the set of cutoff frequency corresponding to each neuron in $S_n^l$ as

$$F_c^l = \{f_c^{n_1^l}, f_c^{n_2^l}, f_c^{n_3^l}, ..., f_c^{n_{\lambda_l}^l}\} \tag{26}$$

Since neurons in $S_n^l$ can have different neuron parameters, from Lemma 3, it is possible to set the parameters such that all entries of $F_c^l$ are distinct. Hence, there exists a permutation $\pi$ such that

$$f_c^{n_{\pi(1)}^l} < f_c^{n_{\pi(2)}^l} < f_c^{n_{\pi(3)}^l} ... < f_c^{n_{\pi(\lambda_l)}^l} \tag{27}$$

Denote the input spike frequency to layer $l$ as $f_{in}^l$, neuron $n_i^l$ is a part of a valid memory pathway in the network if

$$f_c^{n_i^l} \leq f_{in}^l \tag{28}$$

Therefore, for input spike frequency $f_{in}^l \geq f_c^{n_{\pi(\lambda_l)}^l}$, all neurons in $S_n^l$ can be part of a valid memory pathway of the network. For input spike frequency such that $f_c^{n_{\pi(i)}^l} \leq f_{in}^l < f_c^{n_{\pi(i+1)}^l}$, $i$ neurons can be part of a valid memory pathway of the network.

For any input to the network $f_{in} \in [F_{min}, F_{max}]$, denote the number of ways different neurons in $S_n^l$ can be part of valid memory pathways as $k_l$. The total number of distinct memory pathways $\mathcal{K}$ in the network is the product of $k_l$ of all layers:

$$\mathcal{K} = \prod_{l=1}^{m} k_l \tag{29}$$

Since for all layers, $0 \leq k_l \leq \lambda_l$,

$$\mathcal{K} \leq \prod_{l=1}^{m} \lambda_l \tag{30}$$

Consider that, for any layer $l \in \{1, 2, 3, ..., m\}$, the input $f_{in}^l$ is bounded by $[F_{min}^l, F_{max}^l]$. $k_l = \lambda_l$ can be achieved by setting $f_c^{n_{\pi(1)}^l}$ and $f_c^{n_{\pi(\lambda_l)}^l}$ such that:

$$f_c^{n_{\pi(1)}^l} = F_{min}^l \text{ and } f_c^{n_{\pi(\lambda_l)}^l} = F_{max}^l \tag{31}$$

Hence the bound is tight for (30). The proof is complete.

**Lemma 5** For an mMND network with $m$ layers and $\{\lambda_1, \lambda_2, ...\lambda_m\}$ different neuron dynamics in each layer and a skip-layer connection made between layer $l_a$ and $l_b$, s.t. $a, b \in \{1, 2, ...m\}$ and $(b - a) > 1$, the least upper bound of the number of memory pathways is $\prod_{i=1}^{m} \lambda_i + (\prod_{i=1}^{a} \lambda_i \cdot \prod_{i=b}^{m} \lambda_i)$

*Proof.* Denote the mMND network with skip-layer connection between layer $l_a$ and layer $l_b$ as $P$. Denote the set of all neurons in $P$ as

$$S_P = \{n_1^1, n_2^1, n_3^1, ..., n_1^2, n_2^2, n_3^2, ...\} \tag{32}$$

where $n_i^1$ is a neuron in layer $l_1$, and $n_i^2$ is a neuron in layer $l_2$, etc. The activation state $o_{n_i^j}$ of a neuron can be denoted with binary values 0 and 1 with $o_{n_i^j} = 1$ representing that $n_i^j$ receives input frequency that is above its cutoff frequency $f_c^{n_i^j}$.

The set of all possible neuron activation states $O$ in $P$ that generates non-zero network output feature vector can be partitioned into two subsets denoted as $O_A$ and $O_B$.

Set $O_A$ contains all states where the input frequency $f_{in}^k$ to any layer $l_k$ such that $a < k < b$ satisfies

$$f_{in}^k < f_c^{n_i^k} \ \ \forall i \in \{1, 2, 3, ..., \lambda_k\} \tag{33}$$

Set $O_B$ contains all the remaining neuron activation states in $O$, where all layers receive input frequency higher than cutoff frequency of at least one neuron in that layer.

For all the states in $O_A$, no spike signal is sent from layer $b-1$ to layer $b$, since at least one layer between $l_a$ and $l_b$ generates no output. Hence, output from $P$ is not affected if connections between layer $l_i$ and $l_{i+1}$, such that $i \in \{a, a+1, ..., b-1\}$, are removed. Network $P$ is therefore equivalent to network $P'$ that has layers $\{l_1, l_2, ..., l_a, l_b, l_{b+1}, ..., l_m\}$ connected sequentially.

According to Lemma 4, it can be derived that the least upper bound of the number of distinct memory pathways in $P'$ is $\prod_{i=1}^a \lambda_i \cdot \prod_{i=b}^m \lambda_i$.

Hence, for all states in set $O_A$, the least upper bound of the number of distinct memory pathways in $P$ is also $\prod_{i=1}^a \lambda_i \cdot \prod_{i=b}^m \lambda_i$. For all states in set $O_B$, since the activation of neurons in the source layer of the skip-layer connection is already accounted for when considering layer $l_a$, the least upper bound of the number of distinct memory pathways is the same as network $P$ that has no skip-layer connection, which is $\prod_{i=1}^m \lambda_i$ according to Lemma 4.

For the set of memory pathways from states in $O_A$, denoted as $M_A$, and the set of memory pathways from states in $O_B$, denoted as $M_B$, the number of memory pathways of network $P$ is $|M_A \cup M_B|$.

From Lemma 5, the bound is tight for $|M_A| \leq \prod_{i=1}^a \lambda_i \cdot \prod_{i=b}^m \lambda_i$ and for $|M_B| \leq \prod_{i=1}^m \lambda_i$. It also satisfies that $M_A \cap M_B = \emptyset$ since all elements in $M_A$ have $(m - (b - a - 1))$ layers and all elements in $M_B$ have $m$ layers. Hence the bound is tight for

$$|M_A \cup M_B| \leq \prod_{i=1}^m \lambda_i + \left( \prod_{i=1}^a \lambda_i \cdot \prod_{i=b}^m \lambda_i \right) \tag{34}$$

The proof is complete.

## E  Time-varying Function Approximation

A time-varying function f($t$) can be approximated using piece-wise constant function, such as illustrated in Figure 4. It is therefore possible to approximate the time-varying function with a feedforward SNN by approximating each of the constant function with a memory pathway. In this section, we test the approximation capability of feedforward SNN for time-varying functions using this principle. The target functions to approximate have the form of:

$$f(x) = \frac{x^n}{x - m} \tag{35}$$

Here, $x$ is variable; $n$ and $m$ are function parameters. For discrete-time simulation of the network, we approximate the target function with

$$x = t_{in} \ \ \text{and} \ \ f(x) = t_{out} \tag{36}$$

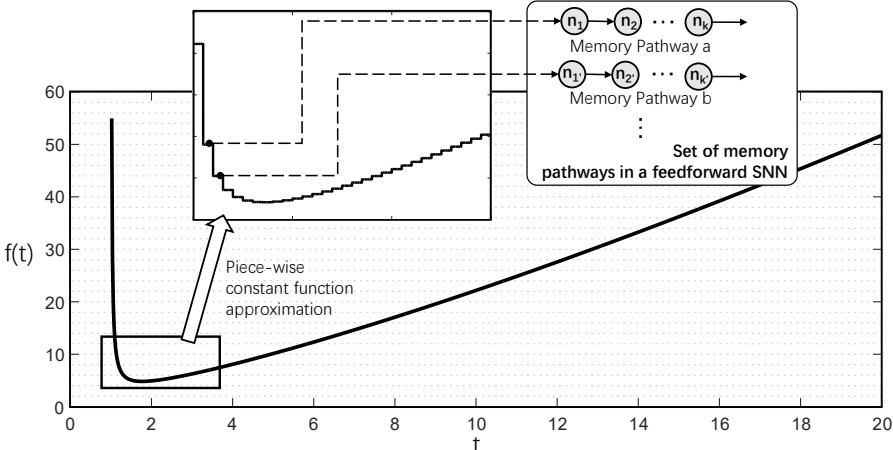

Figure 4: Using a set of memory pathways to map a piece-wise constant function for approximating a time-varying function.

where $t_{in}$ is the input spike period and $t_{out}$ is the output spike period. We test approximation performance of a small-scale feedforward SNN with 6 fully-connected layers, skip-layer connections $\{(2,5),(3,5)\}$ and 4 neuron dynamics. The network is trained with BPTT to minimize mean squared error (MSE) loss between the spike period of network output $t'_{out}$ and target spike period $t_{out}$.

To evaluate the network's approximation performance, we construct 6 networks with the same structure as discussed above, and change each to have different trainable parameter numbers by scaling the layer size. For baseline comparison, networks with 6 layers, no skip-layer connection and using homogeneous neurons, configured to have the same trainable parameter numbers as the proposed networks, are tested. All networks are trained with BPTT. The loss function measures the difference between the network output spike trains and the target spike trains with MSE. Two sets of function parameters: $(m = 1, n = 3.3)$ and $(m = 2, n = 2.1)$ are tested for the target functions $f(x)$ on domain $[3, 10]$. The resulting MSE losses for different network scales are shown at the bottom of Figure 5. It can be observed that the proposed networks can approximate target functions with less error than the baseline networks at all network scales. The smallest tested networks have relatively high losses while performance increases quickly with more trainable parameters. The rate of performance improvement decreases when trainable parameter number is above 4000.

To understand the impact of the target function parameters to approximation performance of SNN, we test baseline and the proposed network with 4167 trainable parameters for different pairs of

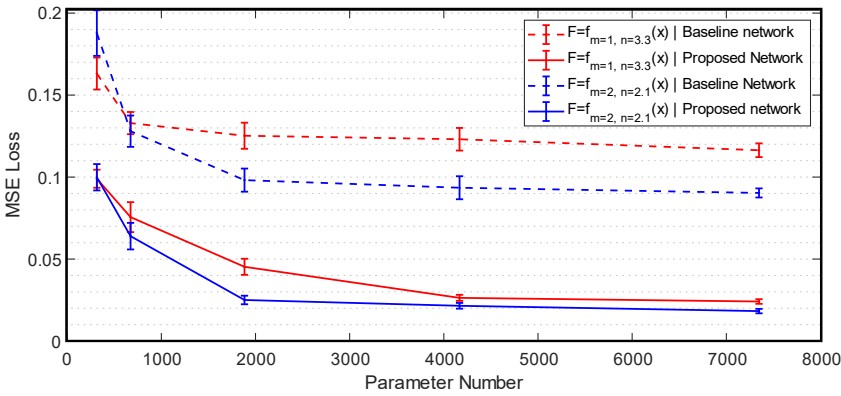

Figure 5: MSE loss vs. number of trainable parameters for the function approximation experiments.

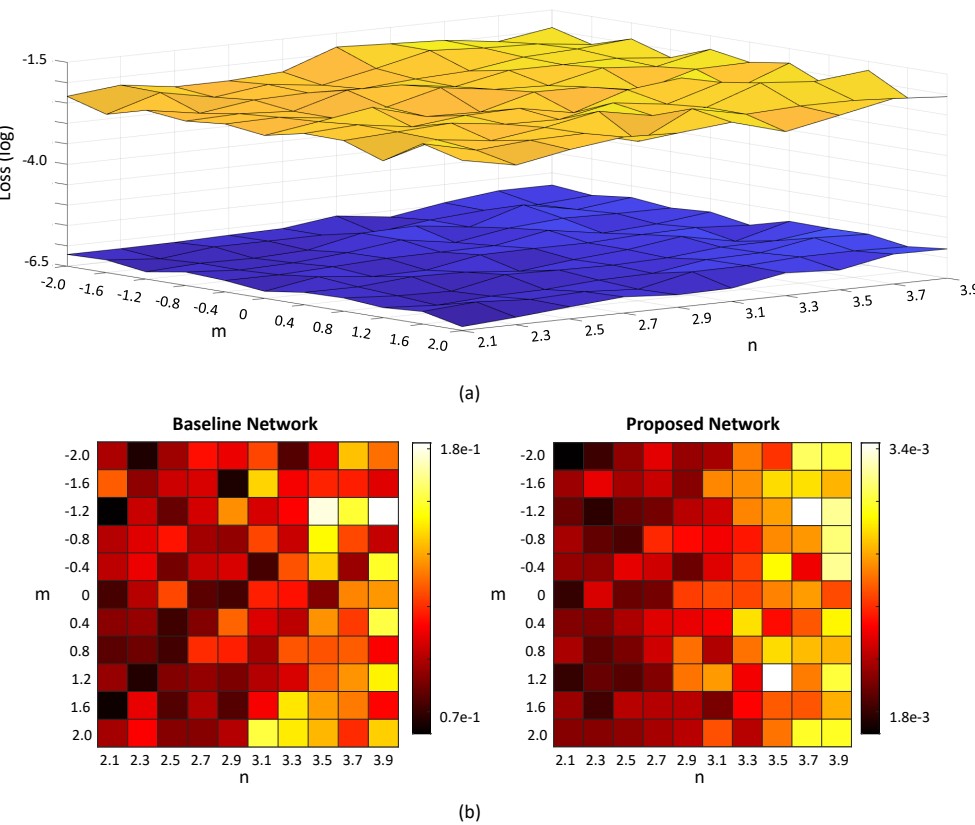

Figure 6: (a) MSE loss (log) of baseline network (top) and the proposed network (bottom) for approximating functions with different parameters $m$ and $n$. (b) Heat plots of MSE loss for approximating functions with different parameters $m$ and $n$.

function parameters $m$ and $n$. The resulting MSE loss is shown with a heat map in Figure 6. It can be observed that for all $m$ and $n$ value pairs, the proposed network achieves lower loss than the baseline network. Another observation is that, there is no clear correlation between the value of $m$ and approximation error. On the other hand, for higher values of $n$, the approximation error generally increases for both baseline and the proposed network.

## F  NETWORK SPIKING ACTIVITY

To investigate neuron spiking activity in the proposed network, for the function approximation task as described in Appendix E, with parameters $(m = 1, n = 2.3)$ and a randomly selected approximation point $f(x = 19)$, we plot the timing of neuron spikes at the last layer over training epochs in Figure 7 (a). It can be observed that the network initially generates spikes at widely distributed timings. After the first 50 training epochs spike timing starts to converge and remains stable at around 200 epochs. The final output spike period is $t'_{out} = 48$, which matches the target output period.

Next, we investigate spiking activity of the BPTT trained network as described in Section 5.4 for N-Caltech101 classification task. Three different test data points are presented to the network, and the spikes from neurons in the first depth of layer 8 and neurons in the final layer, are recorded and shown in Figure 7 (b). Here, (i), (iii) and (v) are from layer 8; (ii), (iv) and (vi) are from the final layer. (i) and (ii) are from the observation of a data point with class label "5"; (iii) and (iv) are from the observation of another data point also with class label "5"; (iii) and (iv) are from the observation of a data point with class label "1".

It can be observed that neurons in layer 8 exhibit similar activity in (i) and (iii) as the network is presented with data points from the same class. (ii) and (iv) show that most spiking activities are

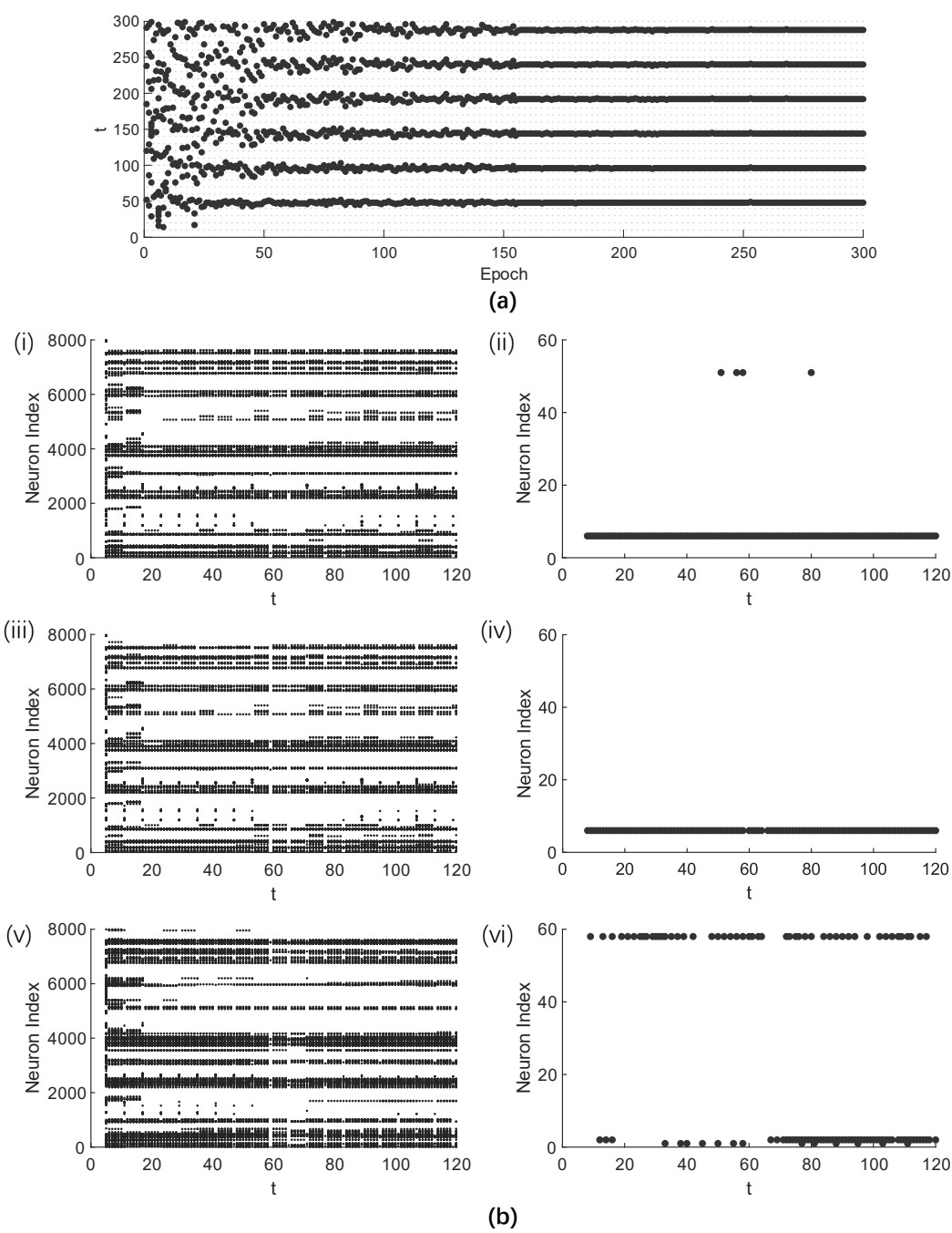

Figure 7: (a) Raster plot for function approximation experiment; $t$ is the simulation time step. (b) Raster plots for the proposed SNN trained for N-Caltech101 with BPTT; $t$ is the simulation time step; (i), (iii) and (v) are from layer 8; (ii), (iv) and (vi) are from the final layer.

from the neuron with index 5, indicating correct prediction for the two data points. When a data point from a different class is presented, spiking activity in (v) shows different patterns than those in (i) and (iii). In the final layer, neuron with index 1 generates the most spikes, leading to a correct prediction. However, for this data point, neuron with index 58 is also generating a considerable number of spikes, indicating that the network is more likely to mis-classify this data point, compared to the other two tested data points.

## G  IMPLEMENTATION OF HETEROGENEOUS CONV-SNN

Multi-neuron-dynamic (MND) networks with convolutional layers can be considered a scaled-up version of the mMND network, where each neuron dynamic now contains a matrix of neurons that receive input features from different spatial locations. To implement heterogeneous Conv-SNN, first consider a regular Conv-SNN layer with no heterogeneity. The spiking neuron matrix has dimension $\{W, H, D\}$, where $D$ is the depth of the layer. The convolution filter has dimension $\{C, w, h, D\}$ where $C$ is the number of input channels and $D$ is the number of output channels.

Based on this, a heterogeneous layer $l$ can be constructed, by concatenating heterogeneous neuron matrices with the same $W$ and $H$ along layer depth. The resulting spiking neuron matrix has dimension $\{W, H, \lambda D\}$, where $\lambda$ is the number of neuron dynamics. Hence, layer depth $i \in \{k+1, k+2, k+3, ..., k+D\}$ have the same neuron parameters, where $0 \leq k \leq \lambda - 1$ is an integer representing the index of neuron dynamics. The convolution filter has the same dimension as the regular Conv-SNN layer: $\{C, w, h, D\}$, which is shared by layer depth with different $k$ values. During forward pass of this layer, a convolution operation is applied to generate an input signal matrix with dimension $\{W, H, D\}$. For each value of $k$, neurons in depth $\{k+1, k+2, k+3, ..., k+d\}$ are simulated based on the neuron parameters for such neuron dynamic index $k$ using the signal matrix as input. For the next convolutional layer $l'$ receiving input from layer $l$, the filter dimension is $\{\lambda D, w', h', D'\}$ with $\lambda D$ input channels.

## H  DETAILS ON THE BAYESIAN OPTIMIZATION PROCESS

During the first stage of the dual-search-space optimization process, the parameters to optimize include: $N_{layer}$, $L_{start}$, $L_{end}$, $N_{skip}$, $N_{dynamic}$, all of which are positive integers. Specifically, $N_{layer}$ is the number of convolutional layers. For skip-layer connection, there are three configuration parameters to optimize: starting layer $L_{start}$, which is the source layer of the first skip-layer connection; ending layer $L_{end}$, which is the target layer of the last skip-layer connection; skip-layer connection number $N_{skip}$, which defines how many connections to implement. The source layer of the $N_{skip}$ skip-layer connections are placed evenly between $L_{start}$ and $L_{end}$, each with skip length equal to

$$\lfloor \frac{L_{end} - L_{start}}{N_{skip}} \rfloor$$

In the case of

$$\frac{L_{end} - L_{start}}{N_{skip}} \neq \lfloor \frac{L_{end} - L_{start}}{N_{skip}} \rfloor$$

,

the value of $L_{end}$ is reduced to the maximum value that satisfies

$$\frac{L_{end} - L_{start}}{N_{skip}} = \lfloor \frac{L_{end} - L_{start}}{N_{skip}} \rfloor$$

For heterogeneity, the number of different dynamic $N_{dynamic}$ in all layers is optimized jointly. The constraints for the parameters are defined as:

$$N_{layer} \in [4, 15]$$

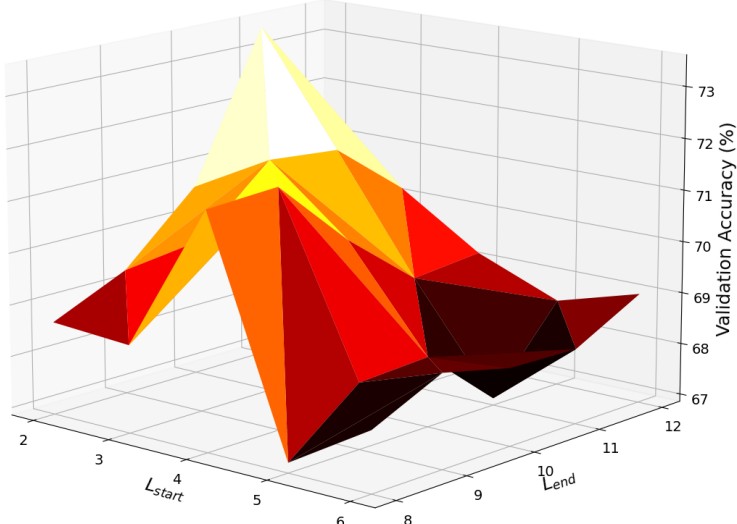

Figure 8: Visualization of the learned distribution from Bayesian optimization: N-Caltech101 validation accuracy vs. skip-layer configurations.

$$2 \leq L_{start} < (N_{layer} - 1)$$
$$(L_{start} + 1) < L_{end} \leq N_{layer}$$
$$0 \leq N_{skip} \leq (L_{end} - L_{start})/2$$

and

$$N_{dynamic} \in [1, 10]$$

The manually configured neuron parameters are, $\tau_m = 100$ and $R_m = 300$ for all neuron dynamics, and $a \in [-30, -5]$ is distributed evenly for each neuron dynamic. Figure 8 shows visualization of the learned distribution from the Bayesian optimization process. Here, mapping from the search space of skip-layer connection configurations to the validation accuracy of N-Caltech101 is shown. The highest validation accuracy is reached at $L_{start} = 2$ and $L_{end} = 11$.

Due to the exponential increase of search space with the number of neuron dynamics in each layer, it is highly inefficient to search for every neuron parameters of each dynamic. In the second stage of the optimization process, we choose to apply Bayesian optimization for the parameter $a$ of each neuron dynamic separately, while $\tau_m$ and $R_m$ are optimized jointly with the same values shared by all neuron dynamics. $a$ values are taken to the precision of $10^0$, and $\tau_m$ and $R_m$ values are taken to the precision of $10^1$. The constraints are $a \in [-30, -5]$, $\tau_m \in [50, 200]$ and $R_m \in [200, 400]$. The value of $t_{nd}$ for all networks is set to 1. The parameters of each optimized network are shown in Table 1. Note, the skip-layer connections are listed as source and target layer pairs.

