# OpenReview forum: "Sequence Approximation using Feedforward Spiking Neural Network for Spatiotemporal Learning: Theory and Optimization Methods"
_ICLR.cc/2022/Conference — ICLR 2022 Poster_

### Official Review · Reviewer_himw · 2021-10-23

**Correctness:** 3
**Technical Novelty And Significance:** 3
**Empirical Novelty And Significance:** 2
**Recommendation:** 6
**Confidence:** 4

**Main Review:**

Theoretical understanding of SNNs in terms of approximation, temporal memorability, optimization, and generalization is a fundamental and challenging problem for SNNs. This paper takes a step in this direction. The motivation is interesting and significant.

Despite the interest of the overall goal of the research, this paper suffers from several limitations,  which make it improper for acceptance. There are two main reasons for rejection, (1) the writing of this paper, especially theoretical conclusions,  needs to be improved for proofreading; (2) the claimed contributions cannot be confirmed or supported by the theory and experiments.

In detail,
1. Definitions 1-5 are too redundant. It's not necessary to state these informal definitions in a formal environment. Please be clear and simple.
2. What's the difference between lemmas and theorems presented by this paper? Is it just because the conclusions of the theorems are more important or more dazzling than those of the lemmas?
3. Can the authors provide a formal description and proof for their theoretical results? The current can only be called a heuristic description.
4. What's G in Eq. (2)? Is it trainable in practice? It works just like a normalizer, and thus, can be replaced by $R_m / \tau_m e^{-1/ \tau_m}$? In fact, the legality of rate coding has already been discussed.
5. Besides, there have been great efforts on the theoretical understanding of SNNs, such as [1-4].
6. Notice that lemmas 4 and 5 only investigate the upper bound, not the least upper bound. So one cannot conclude that using heterogeneous and skip connections can improve the sequence approximation of feedforward SNNs.

The experiment is intriguing and could lead to something new. However, I am unable to follow the author's suggestions, such as how to build an experiment based on past beliefs. Furthermore, don't the optimization techniques discussed in Section 4 already exist? Both BPTT and STDP are trivial for skip connection and feedforward architecture [5]. In order to show the effectiveness of the proposed network, it is not sufficient to use only accuracy as the evaluation indicator. It's better to display the spot rasters of the memory and learner modules, just like what shows in [2] and [6].

I felt a sense of fragmentation when reading this paper. It is very much like a suture monster. I would suggest that the authors use rigorous proof to re-write clearly what the conclusions of the theory are? Where is the proof of innovation? And explain clearly what is the innovation of the experimental model or algorithm? Why is it effective?

[1] On the Algorithmic Power of Spiking Neural Networks. 2019.
[2] Bifurcation Spiking Neural Network. 2019.
[3] Firing rate predictions in optimal balanced networks. 2013.
[4] Spiking Analog VLSI Neuron Assemblies as Constraint Satisfaction Problem Solvers.2015.
[5] Error-Backpropagation in Temporally Encoded Networks of Spiking Neurons. 2000.
[6] Slayer: Spike layer error reassignment in time. 2018.

**Summary Of The Paper:**

This paper presents some theoretical understanding of sequence approximation using feedforward SNNs. The main conclusions are two folds: (1) a feedforward SNN with one neuron per layer and skip-layer connections can approximate the mapping rate-coding function; (2) a feedforward SNN constructed by heterogeneous neurons with varying dynamics and skip-connections can improve sequence approximation. Besides, the authors provide the DSBO to optimize the architecture and parameters of their proposed SNNs.



**Summary Of The Review:**

In summary, I find the social impact of this work, but it still needs to be refined carefully. If there were an option for 'Weak Reject', I would select that. Since there is no such option, I recommend rejection given the many suggestions.

______________
After reading the rebuttal and other reviewers' comments, I consider arising my score to 6, although there are still many inappropriate points in the current version.

---

> ### Author Response · Authors · 2021-11-23
> **Response to Reviewer himw Part 1**
>
> Thank you for your feedback. We reply to each of your comments below.
>
> > “Definitions 1-5 are too redundant. It's not necessary to state these informal definitions in a formal environment. Please be clear and simple.”
>
>
> Reply: Thank you for this suggestion. We have removed the definition of skip-layer connections and neuron delay and explained them in the notations. We decided to keep three definitions for readers not familiar with those terms, but have simplified the definition of memory pathways.
>
>
> > “What's the difference between lemmas and theorems presented by this paper? Is it just because the conclusions of the theorems are more important or more dazzling than those of the lemmas?”
>
> Reply: We arranged the naming of the theoretical development in the paper such that the main topic, which proves the function approximation capability of SNN with skip-layer connection, is named Theorem 1. Therefore, the conclusion of Theorem 1 is more important in the overall context of the paper.
> Lemmas are used to help prove Theorem 1 and evaluate its impact on network design. In particular, proof of Theorem 1 requires the ability to configure a neuron which can have $\gamma$ as any positive integers by changing G. This ability is proved by Lemma 1.
> Lemma 2 is developed to bring out the importance of having skip-layer connections in function approximation. It can be considered as supplementary to Theorem 1.
>
> Besides function approximation, we also provide the theoretical basis for improving SNN’s function approximation using skip-layer connections and heterogeneity with Lemma 3, 4 and 5. They are also important developments that support the proposed network design, but as they are not directly related to proving the networks’ ability for function approximation, we name them as Lemmas.
>
> > “Can the authors provide a formal description and proof for their theoretical results? The current can only be called a heuristic description.”
>
> Reply: Please note that in the main paper we only provided a sketch of the overall proof, and the detailed formal proofs are available in the Appendix. We have also rewritten the proof sections in the Appendix to be more formal.
>
> > “What's G in Eq. (2)? Is it trainable in practice? It works just like a normalizer, and thus, can be replaced by...”
>
> Reply:  $G$ is the synaptic conductance that is trained with STDP or BPTT, thus is not a normalizer. The STDP algorithm used in this work is a common variant that has equations:
>
> $$
> \Delta G_{p} = \alpha_{p}e^{-\beta_{p}({G-G_{min}})/({G_{max}-G_{min}})} \textbf{}
> $$
>
> $$
> \Delta G_{d} = \alpha_{d}e^{-\beta_{d}({G_{max}-G})/({G_{max}-G_{min}})} \textbf{}
> $$
>
> In BPTT we used SGD to train $G$ as network weights.
>
> While rate encoding has been well studied, in this work Lemma 1 is derived to show that it is possible to set neuron response rate $\gamma$ to a certain value. The ability to do so is required in the development of the proof of the sequence approximation capability of SNN (Theorem 1).
>
> > “Besides, there have been great efforts on the theoretical understanding of SNNs, such as [1-4].”
>
> Reply: We agree with the reviewer that there are many prior works on providing better theoretical understanding of SNN including the as mentioned works and works already discussed in Section 2. We also would like to point out that the specific problem of function approximation using feedforward LIF network is different from the topics of those prior works. In particular, [1] and [3] consider SNN for optimization problems, and [2] and [4]  focus on analyzing the dynamic of SNN. We have included the as mentioned works and discussed the difference of this work compared to them in Section 2.

---

> > ### Author Response · Authors · 2021-11-23
> > **Response to Reviewer himw Part 2**
> >
> > > “Notice that lemmas 4 and 5 only investigate the upper bound, not the least upper bound. So one cannot conclude that using heterogeneous and skip connections can improve the sequence approximation of feedforward SNNs.”
> >
> > Reply: Thank you for bringing out this important part. The statements and proofs of  Lemma 4 and Lemma 5 are revised to investigate the “least upper bound” and show that heterogeneous and skip-layer connections increases the least upper bound of the memory pathways and hence, improve the sequence approximation of feedforward SNNs.
> >
> > > “The experiment is intriguing and could lead to something new. However, I am unable to follow the author's suggestions, such as how to build an experiment based on past beliefs. Furthermore, don't the optimization techniques discussed in Section 4 already exist?
> >
> > Reply:  The objective of section 4 is to show that incorporating the concepts of heterogeneity and skip-layer connection within a complex SNN can improve its performance. However, in a multi-layer SNN there are many options for choosing neuronal parameters (heterogeneity) and skip-layer connections. Therefore, section 4 discussed an automated optimization process to tune these parameters aiming to maximize the network’s performance.
> >
> > As the reviewer correctly pointed out, Bayesian optimization is a well-known technique and has also been applied in prior papers to tune SNN parameters, which we acknowledge in Section 2. We don’t consider the BO itself as a contribution of the paper, and revised the text to ensure that BO algorithm itself is not presented as an algorithm.
> >
> > Rather, we focus on designing the dual-search-space option of the BO in the context of the SNN network/parameter optimization and analyzing key observations made in that context. One of the key observations from Section 5 is that single-stage BO where both neuron dynamics and skip-layer connections are tuned simultaneously has a slower convergence than tuning them sequentially during two stages. Another interesting observation is that optimizing dynamics and optimizing network structures have different impacts on BPTT and STDP trained networks.
> >
> >
> >
> >
> > > “In order to show the effectiveness of the proposed network, it is not sufficient to use only accuracy as the evaluation indicator. It's better to display the spot rasters of the memory and learner modules, just like what shows in [2] and [6].”
> >
> > Reply: We have included an additional function approximation experiment in Appendix E, where we compare the proposed design and baseline networks for different network scales, and observe that the proposed networks have better function approximation performance.
> >
> > We included a new Appendix F which presents raster plots for both the network used in function approximation test and the network used in N-Caltech101 classification.
> >
> > > “Both BPTT and STDP are trivial for skip connection and feedforward architecture [5].”
> >
> > We agree with the reviewer, and therefore, the paper does not claim that training the proposed SNN with BPTT and STDP as new techniques. Rather it stresses on the fact that the proposed architecture with heterogeneity and skip-layer connection, shows improved performance over baseline (homogeneous neurons without skip-layer) irrespective of whether the network is trained with STDP or BPTT. Figure 2 and Section 4 are presented primarily to demonstrate how one can realize heterogeneous neuron dynamics and skip-layer connections for Conv-SNN or MLP-SNN architecture, as well as the changes that are necessary to train them. We believe these sections provide the details necessary to understand and reproduce the accuracy results. There are some nuances, such as the use of transferred and learned synapses, as well as the separation of learner and memory modules, that are not common in prior works.

---

> > > ### Comment · Reviewer_himw · 2021-11-23
> > > **Reponse to the Rebuttal.**
> > >
> > > I have read the revised paper and believe that there is a great improvement, in which the theorems and lemmas are more rigorous and the corresponding proofs are more formal and easier to proofread. In the current version, the contributions of this paper will be better exhibited. So I tend to accept this paper and arise my score to 6, although there are still many inappropriate points in the current version.
> > >
> > > The following lists some suggestions:
> > > - It is better to specify the relation between Eq. (1) and Eq. (2) by clearly invoking $I(t) =  \mathbf{G} \sum_{i} \delta\left( t-t^i \right)$.
> > > - Or provide an appropriate reference, e.g., [1],  to indicate the source of Eq. (2).
> > > - Many symbols are redundant.
> > > - The manuscript has areas, especially the description of lemmas and proofs, with awkward phrasing and some grammatical errors throughout and should be proofread. For example, what's the spike period function $P(t)$ in Theorem 1? Is this mentioned above? Besides, I do not agree that the proof for the least upper bound in Lemma 4 is reasonable. It is not trivial to show the minimum of  the upper bound of the path number.
> > >
> > > Anyway, the authors have also struggled a lot of efforts, so be it, I hope to see a better camera ready, if luckily accepted.
> > >
> > > [1] Time structure of the activity in neural network models. 1995.

---

> > > > ### Author Response · Authors · 2021-11-29
> > > > **Reply to reviewer himw**
> > > >
> > > > Thank you for your positive feedback. We address each of your comments below:
> > > >
> > > > > It is better to specify the relation between Eq. (1) and Eq. (2) by clearly … Or provide an appropriate reference, ...
> > > >
> > > > Reply: We will add this to the final version if the paper is accepted.
> > > >
> > > > > Many symbols are redundant.
> > > >
> > > > Reply: We will more carefully proofread the paper and remove any redundant symbols.
> > > >
> > > > > The manuscript has areas, especially the description of lemmas and proofs, with awkward phrasing and some grammatical errors throughout and should be proofread. For example, what's the spike period function $P(t)$ in Theorem 1? Is this mentioned above?
> > > >
> > > > Reply: We will more carefully proofread the paper.
> > > > $P(t)$ is the function that maps input spike period to output spike period by the network. We will add the definition of $P(t)$ above Theorem 1 in the final version.
> > > >
> > > > > Besides, I do not agree that the proof for the least upper bound in Lemma 4 is reasonable. It is not trivial to show the minimum of the upper bound of the path number.
> > > >
> > > > Reply: We will improve the clarity of proof for Lemma 4. We would like to point out that, in the proof of Lemma 4, for layer $l$ and input spike sequence with frequency $f_{in}^{l}$, consider the permutation $\pi$ as described in the proof: there exists an unique value $k$ such that only neurons with index $\pi(i) \text{  s.t.  }  i \leq k$ can spike. This is because if neuron $\pi(k)$ receives input spike sequence with frequency higher than $f_{c}^{n_{\pi(k)}^{l}}$, then $f_{in}^{l}>f_{c}$ holds for all neurons in layer $l$ with lower cut-off frequency than $\pi(k)$. This limits the number of all possible combinations of neurons in layer $l$ that receive input frequency higher than their cut-off frequency to the number of possible values $k$ can take, which is $\lambda_l$. Throughout all layers, the maximum number of memory pathways thus cannot exceed the product of the number of all such possible combinations in each layer, which is thus the least upper bound in Lemma 4: $\prod_{i=1}^{m} \lambda_{i}$. We will improve the proof of Lemma 4 according to your comment and this response.

---

### Official Review · Reviewer_EDAS · 2021-11-03

**Correctness:** 3
**Technical Novelty And Significance:** 3
**Empirical Novelty And Significance:** 3
**Recommendation:** 6
**Confidence:** 3

**Main Review:**

Pros:
* very good accuracy

Cons:
* many doubts need to be cleared about the theory
* the connection between the theory and the experiments is unclear

The theory is unclear:
* Does it only apply to sequences with constant inter-spike intervals (ISI), as Fig 1a suggests? The fact that they focus on the spike counts (gamma) also suggests that.
If true, this is a strong loss of generality, which should be properly acknowledged. Also, I don't understand how the theory can help with DVS datasets, in which the ISI are highly variable.
If false, then Fig 1 should show variable ISIs, and the authors should justify why they can summarize a spike train by its spike count.
* Lemma 1 seems wrong: what if no input spike? Then changing G has no effect on the response...
* Does the theory work in continuous or discrete-time? The equations suggest continuous time. But a timestep is mentioned on page 4. Plus BPTT works in discrete-time.

The connection between the theory and the experiments is not clear to me. Does the theory only suggests heterogeneous neurons and skip connections, but then the training is done with normal BPTT (or STDP)?
Does the theory also justify the use of learned and transferred synapses? BTW, these concepts are obscure to me. Isn't it equivalent to having a set of weights that is shared between neurons (like in a convolutional layer)?


MINOR POINTS:
* Eq 1: you should say that a is the resting potential
* Eq 2: don't use t for the integration variable
* Table 1 should specify the number of timesteps
* Table 3 should include: https://arxiv.org/abs/2007.05785


**Summary Of The Paper:**

The authors introduce a theory on how to learn arbitrary input spike train - output spike train mappings, using a feedforward SNN with heterogeneous neurons and skip connections.
Then this theory is used to train a deep SNN using either BPTT or STDP. The SNN is then evaluated on IBM DVS Gesture, N-Caltech101. The accuracy that they get with BPTT is beyond the SOTA.

**Summary Of The Review:**

Potentially interesting given the accuracy, but unclear

---

> ### Author Response · Authors · 2021-11-17
> **Response to Reviewer EDAS Part 1**
>
> Thank you for your feedback. We reply to each of your comments below.
>
> > “Does it only apply to sequences with constant inter-spike intervals (ISI), as Fig 1a suggests? The fact that they focus on the spike counts (gamma) also suggests that. If true, this is a strong loss of generality, which should be properly acknowledged. Also, I don't understand how the theory can help with DVS datasets, in which the ISI are highly variable. If false, then Fig 1 should show variable ISIs, and the authors should justify why they can summarize a spike train by its spike count.”
>
> Reply: Since functions with time-varying ISI on a compact interval can be approximated using piece-wise constant functions and one memory pathway maps one constant ISI, a network that contains multiple memory pathways that can respond to multiple constant ISI and thus approximate time-varying ISI. We will modify Figure 1 in the revised paper to reflect the network’s response to time-varying ISI.
>
> The underlying theoretical question is, whether there exists a feedforward SNN that can achieve mapping of any arbitrary ISI. We prove with Theorem 1 that any mapping function from constant ISI to constant ISI on a compact domain can be approximated with a memory pathway. With Lemma 4 and 5 we prove the design of skip-layer connection and heterogeneous neurons can increase the number of distinct memory pathways a network can achieve and thus increase the range of ISI that can be mapped. This leads to the network design process that improves a feedforward SNN’s capability to approximate time-varying ISI mapping function. We show that the networks as implemented in the experiment can therefore be used to process the time-varying input of DVS datasets.
>
>
> > “Lemma 1 seems wrong: what if no input spike? Then changing G has no effect on the response...”
>
> Reply: Lemma 1 is derived for input spike sequences with ISI on a compact range [$T_{min}$, $T_{max}$]. Input sequence with no spike has an infinitely large ISI, thus is not in the compact range. We will clarify this point in the paper by adding below Lemma 1 as following:
>
> “Note, input sequence that has no spike has $t_{in}$ tending to infinity, thus violating the bounded constraint mentioned in Lemma 1.”
>
>
> > “Does the theory work in continuous or discrete-time? The equations suggest continuous time. But a timestep is mentioned on page 4. Plus BPTT works in discrete-time.”
>
> Reply: The theory is developed for the continuous-time dynamic model of SNN. We have changed the text on page 4 to remove any confusion on this point. We would like to point out that the theory focuses on proving the existence of SNN with certain approximation capability and is decoupled from the training process. We train our model with discrete-time simulation, where BPTT can be implemented.
>
> The new text in the revised paper on page 4 is:
>
> “From (2), there exists a value of $u$ such that ...”

---

> > ### Author Response · Authors · 2021-11-17
> > **Response to Reviewer EDAS Part 2**
> >
> > > “The connection between the theory and the experiments is not clear to me. Does the theory only suggests heterogeneous neurons and skip connections, but then the training is done with normal BPTT (or STDP)?”
> >
> > Reply: We would like to point out that the theory developed in this work proves the existence of a feedforward SNN with function approximation capability, and the structural designs that can improve the approximation capability by increasing the range of ISI mapping functions that can be approximated. In other words, the theory suggests heterogeneous neurons and skip-layer connection can be used to improve the network’s function approximation capability. The theoretical results are not related to learning algorithms (BPTT or STDP) used to train the networks’ synaptic conductance.
> >
> > On the other hand, the experimental results demonstrate that when the proceeding constructs are used to optimize a regular convolutional SNN, the resulting SNN architecture can be trained with known training methods (BPTT and STDP) to perform classification on spatiotemporal data. Our experiments compare a baseline convolutional SNN and SNN with the proposed design with simple and complex datasets, and show improved accuracy from the proposed networks, therefore empirically demonstrate the benefit of using heterogeneous neurons and skip-layer connections.
> >
> >
> > > “Does the theory also justify the use of learned and transferred synapses?”
> >
> > Reply: The theory does not suggest the use of learned and transferred synapses. The design of learned and transferred synapses are used to implement the multi-neuron-dynamic (MND) network in a convolutional network structure. This network is a scaled-up version of the mMND network considered in the developed theory.
> >
> >
> > > “BTW, these concepts are obscure to me. Isn't it equivalent to having a set of weights that is shared between neurons (like in a convolutional layer)?”
> >
> > Reply: The transferred synapse is different from weight sharing in regular convolutional SNN in two ways. First, for transferred synapses, the weight is not shared among different spatial locations of the input, but used at the same location. Second, in regular convolutional SNN the shared weight is used to perceive features generated by homogeneous neurons rather than heterogeneous neurons. This design achieves MMD network in an SNN with convolutional layers to increase the upper-bound of distinct memory pathways as discussed by Lemma 4.
> >
> > > “MINOR POINTS: ... ”
> >
> > Reply: We will take the suggested minor points into consideration for the revised paper.

---

> > > ### Comment · Reviewer_EDAS · 2021-11-23
> > > **Concerns addressed**
> > >
> > > My main concerns have been addressed. The theory is now clearer, as well as its connections with the experiments.
> > > I've raised my score to 6 (actually, I would choose 7 if I could).

---

> > > > ### Author Response · Authors · 2021-11-29
> > > > **Reply to reviewer EDAS**
> > > >
> > > > Thank you for your positive feedback.

---

### Official Review · Reviewer_oV8W · 2021-11-06

**Correctness:** 3
**Technical Novelty And Significance:** 3
**Empirical Novelty And Significance:** 3
**Recommendation:** 8
**Confidence:** 4

**Main Review:**

Major Points

The paper is well written, the motivation of the authors is clear, and the framework presented is easy to understand. The theory part is written in a clear mathematical manner, with a theorem-proof structure, which I really like. But I have one main point of criticism that seems very important to me, while the entire theory section provides insights and interesting ideas the provided results in sec. 5 Experiments does not hold up to this. Certainly, a benchmark comparison on common data sets with other methods is very important, but it’s not adequate for the hypotheses presented in the theory part. The main argument of this paper (at least for me) is that a feedforward SNN with many memory pathways can approximate a mapping between a temporal sequence of spike trains with time-varying unknown frequencies to a pre-defined output spike trains with known frequencies, where the number of memory pathways can be increased by optimizing the skipping connections and heterogeneities in the network. This connection to the universal approximation theorem should be more supported by quantitative experiments. My suggestion to investigate the computational power of the optimised SNN would be to see how big the memory effect is as well as to exam the complexity of the mapping the SNN can approximate. One way to address this would be to train the SNN to learn a mapping from $f(x)=x $ to $g(x)=(x^n)/f(x-m)$ for different combinations of $m=1,\dots, M$ and $n=1,\dots, N$, e.g. $M=N=10$.  The resulting reconstruction errors could then be considered as a function of m and n as well as the number of network parameters. If all this should provide very good reconstructions, the mapping function could be chosen in a more complicated design.
A next point: On page 3 “[…] as for MLP-SNN and Conv-SNN network the analysis can be extended according to the specific layer dimensions.” At this point more details or explanations would be desirable I only understood this part after I had a look at the source code.

Page 7: Please provide a formal definition for the objective function, which metric is optimised?

Page 7: “[…] first optimized with fixed, manually selected neuron parameters.” How were these parameters elected or on what basis, I assume random values should not work?

Page 8 Fig 3.: I would suggest using a different symbol for the stage 1 and stage 2 optimisation. Further, I agree with the authors that the dual search converges faster, but it seems that the single search achieves a marginally better validation error. In order to better assess the advantage of faster convergence, how long does an iteration step take and what hardware was used?

Minor Points
Page 5: “Since any continuous bounded function on a compact interval […]”. For the sake of completeness, I would suggest adding the definition for a compact interval. How were these parameter selected?

Page 5 and other use $\left( … \right)$, e.g. Lemma 3

Page 7: This is really optional: Would it be possible to show some projections of the search space from the architecture search space? Maybe in the appendix or even the code-section.

Page 8: It’s a matter of taste: I would suggest that the y axis in figure 3 has the same range in both plots. then the plots can be compared directly with each other without having to pay attention to the axis section.


**Summary Of The Paper:**

In this publication, the authors present an interesting framework for approximation the mapping between sequences by feedforward spiking neural networks (SNN), providing insights on the computational capabilities of feedforward SNNs for the approximation of the mapping of input to output spike trains. And how these are influenced by hyperparameters such as the network architecture, heterogenic properties of neurons and skipping connections. The authors also provide an ansatz how these hyperparameters can be optimised in a two-steps Bayesian optimisation fashion. The performance of the approach is demonstrated by several spatiotemporal classification tasks, on datasets like DVS Gesture, N-Caltech 101 and sequential MNIST.

**Summary Of The Review:**

The paper is well written, the motivation of the authors is clear, and the framework presented is easy to understand. The theory part is written in a clear mathematical manner, with a theorem-proof structure, which I really like. The theoretical results provide new insights and should give new impulses to other scientists in this field as well as in related fields, e.g.: for improved applications as well as building theories.

My main criticism is that the good theoretical results are not sufficiently well supported by numerical experiments, which could further consolidate the good theory results. I hope to provide a possible motivation for supplementing the results in my review. However, these would have to be provided in order to recommend a publication.

---

> ### Author Response · Authors · 2021-11-23
> **Response to Reviewer oV8W Part 1**
>
> Thank you for your feedback. We reply to each of your comments below.
>
>
> > “The paper is well written, the motivation of the authors is clear, and the framework presented is easy to understand. The theory part is written in a clear mathematical manner, with a theorem-proof structure, which I really like. But I have one main point of criticism that seems very important to me, while the entire theory section provides insights and interesting ideas the provided results in sec. 5 Experiments does not hold up to this. Certainly, a benchmark comparison on common data sets with other methods is very important, but it’s not adequate for the hypotheses presented in the theory part. The main argument of this paper (at least for me) is that a feedforward SNN with many memory pathways can approximate a mapping between a temporal sequence of spike trains with time-varying unknown frequencies to a pre-defined output spike trains with known frequencies, where the number of memory pathways can be increased by optimizing the skipping connections and heterogeneities in the network. This connection to the universal approximation theorem should be more supported by quantitative experiments. My suggestion to investigate the computational power of the optimised SNN would be to see how big the memory effect is as well as to exam the complexity of the mapping the SNN can approximate. ....”
>
> Reply: Thank you for the detailed suggestion. We have included the suggested experiment in Appendix E (page 18) of the revised paper. In this new experiment we show that the proposed design of using skip-layer connections and heterogeneity performs better at approximating the time-varying function than networks without such designs. We compare networks with different trainable parameter numbers. We also show how the approximation error changes with parameters $m$ and $n$ of the time-varying function and how accurately the network can approximate it, as you have suggested.
>
> > “A next point: On page 3 “[…] as for MLP-SNN and Conv-SNN network the analysis can be extended according to the specific layer dimensions.” At this point more details or explanations would be desirable I only understood this part after I had a look at the source code.”
>
> Reply: We changed the text to make a clearer explanation for the multi-neuron-dynamic (MMD) implemented with MLP or convolutional layer structures in Section 4, and directed the as mentioned discussion to it. We have added more details to the implementation in Appendix H.
>
> > “Page 7: Please provide a formal definition for the objective function, which metric is optimised?”
>
> Reply: We have improved the discussion on dual-search space Bayesian optimization based on your suggestion.
>
> > “Page 7: “[…] first optimized with fixed, manually selected neuron parameters.” How were these parameters elected or on what basis, I assume random values should not work?”
>
> Reply: By observing the distribution of neuron activations in networks within the pre-defined range of network structure configurations considered in the Bayesian optimization process, the parameters were hand-tuned to ensure neuron activations are within a proper range. The goal of the manual tuning is to prevent spiking neurons from having too high or no spike response to the training data, which makes it difficult for the training algorithm to optimize conductance values.
>
> > “Page 8 Fig 3.: I would suggest using a different symbol for the stage 1 and stage 2 optimisation. Further, I agree with the authors that the dual search converges faster, but it seems that the single search achieves a marginally better validation error. In order to better assess the advantage of faster convergence, how long does an iteration step take and what hardware was used?”
>
> Reply: We have improved Figure 3. The training is done with a Nvidia RTX 2080 Ti and we would like to note that each iteration step takes different time depending on the specific network structure that is being surveyed by the optimization process. We observe that the average time per iteration is around 11 hours, thus the time saving from using dual-search-space Bayesian optimization is considerable.
>
>
> > “Minor Points Page 5: “Since any continuous bounded function on a compact interval […]”. For the sake of completeness, I would suggest adding the definition for a compact interval. How were these parameter selected?”
>
> Reply: The bound of the function is determined by the input we are trying to reconstruct, which, in terms of the experiment, depends on the target dataset. For a time-varying function, it is possible to approximate each small segment with a constant function. Therefore, a continuous bounded function on a compact interval ensures that the range and domain of the target function we are trying to approximate is bounded. We have rewritten this sentence in the revision to clarify this point. We have also added Figure 4 in the appendix to help explain this.

---

> > ### Author Response · Authors · 2021-11-23
> > **Response to Reviewer oV8W Part 2**
> >
> > > “Page 7: This is really optional: Would it be possible to show some projections of the search space from the architecture search space? Maybe in the appendix or even the code-section.”
> >
> > Reply:  Thank you for this suggestion. We are modifying the code to save intermediate states during the Bayesian optimization to plot the projection. We will include it in the final version of the paper if it is accepted.
> >
> >
> > > “Page 8: It’s a matter of taste: I would suggest that the y axis in figure 3 has the same range in both plots. then the plots can be compared directly with each other without having to pay attention to the axis section.”
> >
> > Reply: We have modified the plots based on your suggestion.

---

> > > ### Comment · Reviewer_oV8W · 2021-11-23
> > > **Major points adressed**
> > >
> > > My major points I have outlined have been well addressed. In particular, the additional studies now support the theory sufficiently well. The proposed approach significantly improves the performance.
> > >
> > > But there is one small thing I would like to add. Fig 6.: Please use the same range for the colour bars in both plots, so that the improvement can be seen more clearly.
> > >
> > > Anyhow, I recommend a publication
> > >
> > > Recommendation: 8: accept, good paper

---

> > > > ### Author Response · Authors · 2021-11-29
> > > > **Reply to reviewer oV8W**
> > > >
> > > > Thank you for your positive feedback.
> > > >
> > > > > But there is one small thing I would like to add. Fig 6.: Please use the same range for the colour bars in both plots, so that the improvement can be seen more clearly.
> > > >
> > > > Reply: We will update Figure 6 in the final version to show the improvement more clearly, if the paper is accepted.

---

### Author Response · Authors · 2021-11-23
**Summary of revision**

We would like to thank all reviewers for their helpful feedback. Based on the suggestions from the reviewers, we have made the following changes to the main paper:

* In the Introduction Section's contribution list, item 4 is modified to more accurately describe the contribution.
* More prior works on theoretical studies of SNN are discussed in Section 2.
* Figure 1 (a) is modified to reflect the different response to time-varying functions by different memory pathways. Figure 1 (b) is changed to make the entire figure to fit inside the original space.
* In Section 3.1, definitions of neuron delay and skip-layer connections are moved to notations. The definition of memory pathways is simplified. The discussion on mMND versus MLP-SNN and Conv-SNN is updated.
* In Section 3.2, the definition of $a$ is changed to resting potential.
* The proof sketches are improved.
* In Section 4, the description of network templates are improved. The objective function of Bayesian optimization is defined formally. The description of dual-search-space Bayesian optimization is improved.
* The y-axis limits in Figure 3 are changed to the same values for the two subplots.
* One more baseline is added to Table 3 for comparison.

The changes made to the appendix include:

* All proofs are re-written to be more formal
* The proofs to Lemma 4 and Lemma 5 are developed to prove for the increase of least upper bound.
* A new Appendix E is included which discusses the additional experiments on testing the proposed design for time-varying function approximation. Figure 4, 5 and 6 are added for this section.
* A new Appendix F is included which discusses neuron spiking activities in the proposed networks.
* A new Appendix G is included to describe the details of implementing heterogeneous Conv-SNN
* Format in Appendix H is improved.

---

### Decision · Program_Chairs · 2022-01-20

**Decision:**

Accept (Poster)

**Comment:**

The authors provide a theory for training feed-forward spiking neural networks (SNNs) on input-to-output spike train mappings. They utilise for this heterogenous neurone and skip connections.

The resulting method is tested on DVS Gesture, N-Caltech 101 and sequential MNIST.
It achieved very good performance. The reviewers agreed that the results are interesting and significant.

In the initial reviews, the reviewers pointed out some doubts about the theory and clarity of writing.
These doubts and objections were addressed in the revision and the reviewers were quite satisfied with that.

In conclusion, the manuscript presents interesting results for SNNs with a solid theory and very good experimental results. All reviewers vote for acceptance.